# ChatAni: Language-Driven Multi-Actor Animation Generation in Street Scenes

## Abstract

Generating controllable traffic participant animations from instructions is useful for autonomous driving simulations. Existing methods, however, provide limited support for diverse participants and their dynamic interactions in street scenes. In this paper, we present ChatAni, a language-driven framework for generating interactive and controllable multi-actor animations. To produce fine-grained animations, ChatAni introduces two animators: PedAnimator, a unified multi-task animator that generates interaction-aware and physically plausible pedestrian animations under varying task plans, and VehAnimator, a kinematics-based policy that generates kinematically compliant vehicle animations. To support scene-level control from complex language, ChatAni employs a multi-LLM-agent role-playing approach, using natural language to plan the trajectories and behaviors of different participants. Experiments, including an end-to-end human evaluation over 120 generated scenes, indicate that ChatAni can generate interactive street-scene animations with vehicles and pedestrians, and that the generated data provides measurable gains in the evaluated prediction and understanding tasks. We include key implementation materials, prompts, configuration details, evaluation protocols, and model settings in the supplementary materials, and plan to release the curated full codebase, generated data, and trained checkpoints through a public repository.

## 1 Introduction

Street-scene animation specifies how traffic participants move and interact over time, and is a key component of driving simulation, game and film production, and controllable video generation Caesar et al. (2020); Sun et al. (2020); Xiao et al. (2021); Dosovitskiy et al. (2017). For autonomous-driving simulation, such animation should be controllable, diverse, and consistent with traffic context: vehicles should follow road geometry, pedestrians should move naturally, and different participants should interact through events such as yielding, crossing, hailing, overtaking, or collision avoidance. However, generating such multi-actor animations from user instructions remains challenging because it requires both scene-level planning and low-level executable motion generation.

Traditional driving simulators such as CARLA Dosovitskiy et al. (2017) provide effective manually authored assets and behavior templates. However, when the same templates are repeatedly applied across large-scale scenes, the resulting behaviors can appear rigid or repetitive, and scaling them to diverse, language-conditioned, fine-grained multi-actor interactions requires substantial manual design.

Learning-based methods partially reduce this manual effort, but they often focus on either scene-level planning or low-level motion generation in isolation. At the planning level, LCTGen Tan et al. (2023) and CTG++ Zhong et al. (2023a) use language to generate vehicle trajectories, but they do not model pedestrians and therefore cannot cover pedestrian-involved interactions. At the low level, Pacer Rempe et al. (2023) and Pacer+ Wang et al. (2024) generate fine-grained pedestrian animations, but they do not address map-constrained multi-actor planning or interactions between pedestrians and vehicles.

This creates two coupled challenges. At the planning level, natural-language commands must be converted into map-constrained, temporally consistent, multi-actor plans. At the animation level, these plans must be executed by pedestrian and vehicle controllers without producing physically implausible motion, unstable contacts, or kinematically invalid vehicle states.

To connect language-level scene intent with executable traffic animation, we introduce ChatAni, a language-driven framework for interactive multi-actor street animation generation, as shown in Fig. 1. ChatAni addresses these two levels by producing interaction-aware scene plans and executing them with pedestrian and vehicle animators.

In ChatAni, we design specialized low-level animation generators tailored to the distinct characteristics of pedestrian and vehicle motions. For interaction-aware pedestrian animation, we introduce PedAnimator, a unified multi-task framework that generates physically plausible animations via physics-driven control and diverse input signals. PedAnimator is designed to support physical interactions among pedestrians and between pedestrians and vehicles through an interaction training strategy that integrates general observations and goals, enabling diverse interactions with minimal adjustments. It supports control over both trajectory and body motion, with policy unification achieved

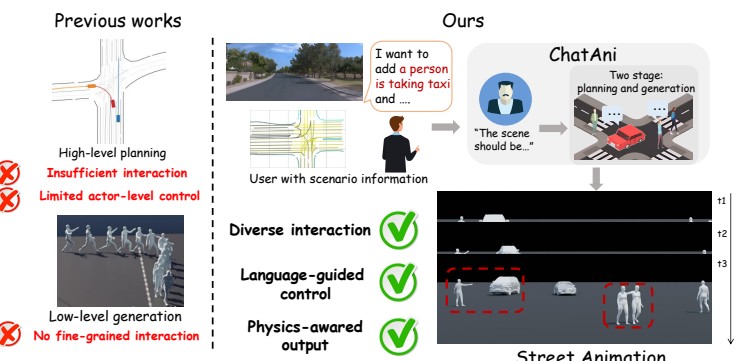

Figure 1: ChatAni synthesizes interactive, multi-actor street animations from natural language by combining LLM-agent-based high-level planning with low-level pedestrian and vehicle animators. The LLM agents produce structured scene and behavior plans, while PedAnimator and VehAnimator drive articulated human bodies and vehicle states to produce the final animations in the evaluated settings. The bottom-right panel shows three consecutive frames $(t_1-t_3)$ from the same generated animation.

through a task-masking mechanism that enables generation under a single policy. Furthermore, hierarchical control and body-masked adversarial motion priors are incorporated, introducing priors into both action and reward spaces. These support multiple control objectives and improve the physical plausibility and human-likeness of the generated animations under our evaluations.

To generate kinematically compliant vehicle animations, we introduce VehAnimator, a kinematics-based control policy for vehicle animation. It converts planned raw trajectories into kinematically compliant animations through a bicycle-model transition, reducing artifacts such as tail swinging and drifting. The framework incorporates obstacle-aware training for collision avoidance in the evaluated settings. History-aware designs aim to improve tracking accuracy and temporal consistency.

To achieve language-driven high-level control and multi-actor interaction planning, ChatAni models each traffic participant as an agent and employs a multi-LLM-agent role-playing approach. Agents organize their own information based on requirements and communicate with other agents, ultimately invoking tools to output plans that are executed by PedAnimator and VehAnimator. Using the language-understanding capability and prior knowledge of LLMs, the design supports structured control over scene-level animation plans through natural language, while inter-agent communication facilitates interaction-aware planning. The LLM agents do not synthesize rendered pixels, body meshes, or fine-grained anatomical details directly; they produce structured high-level plans that are executed by the two low-level animators.

ChatAni is designed around three goals: *interactivity, physical plausibility,* and *controllability.* For interactivity, low-level animators support pedestrian-pedestrian and pedestrian-vehicle interactions, while multi-agent LLM role-playing handles interaction-aware high-level planning. For physical plausibility, PedAnimator employs hierarchical control with body-masked adversarial motion priors to generate human-like motions, while VehAnimator simulates vehicle kinematics. For controllability, LLM agents translate language commands

into structured plans that are executed by the animators. The final output animations are suitable for driving simulators, based on either rendering engines or diffusion models.

Our contributions include: (1) a language-driven framework for multi-actor street scene animation generation; (2) PedAnimator enabling interactive pedestrian animations with varied control signals; (3) VehAnimator creating kinematically compliant animations from trajectories; (4) a multi-LLM-agent framework generating language-controlled, interaction-aware high-level plans; (5) experiments, including an end-to-end human evaluation over 120 generated scenes, showing command alignment, interaction success, user preference, and measurable gains in the evaluated prediction and understanding tasks.

## 2 Related Works

Table 1: Method capabilities comparison

| Method | Traffic Flow Generation | Kinematic Human Motion | Physics-based Human Motion | Traditional Rendering Simulator | **ChatAni** |
|---|---|---|---|---|---|
| Language-controllable | ✓ | ✓ | × | × | ✓ |
| Diversity | ✓ | ✓ | ✓ | × | ✓ |
| Vehicles | ✓ | × | × | ✓ | ✓ |
| Pedestrians | × | ✓ | ✓ | ✓ | ✓ |
| Fine-grained Interaction | × | ✓ | ✓ | ✓ | ✓ |
| Physical Feasibility | × | × | ✓ | ✓ | ✓ |

**Human animation generation.** Human animation generation can be broadly divided into kinematic and physics-based approaches. For kinematics-based generation, transformer-based methods Athanasiou et al. (2022); Guo et al. (2024; 2022) and diffusion models Zhang et al. (2022; 2023) generate motion sequences from text or action descriptions. Recent controllable methods Shafir et al. (2023); Xie et al. (2023); Wan et al. (2023) further introduce conditions such as trajectories, sparse joints, scene positions, or timing constraints. These methods provide flexible semantic and spatial control, but the outputs are usually kinematic pose sequences rather than physics-simulated controllers, so balance, contact forces, external perturbations, and physically reactive interactions are not explicitly enforced. For physics-based generation, existing work such as Peng et al. (2021; 2022); Won et al. (2022); Luo et al. (2023; 2024) learns controllers for locomotion and predefined skills with physically plausible motion. Other methods Tessler et al. (2023); Juravsky et al. (2022); Bae et al. (2023); Xu et al. (2023a;b) extend skill coverage, task composition, or control interfaces. Recent methods such as UniPhys Wu et al. (2025) further support flexible physics-based character control from multimodal inputs such as text, trajectories, and goals. Pacer Rempe et al. (2023) and Pacer+ Wang et al. (2024) focus on physics-based pedestrian animation in street scenes. However, these methods do not directly cover map-constrained multi-actor street animation with both pedestrian-pedestrian and pedestrian-vehicle interactions. Our PedAnimator instead trains a unified physics-based policy that covers following, imitation, and selected interaction tasks.

**Vehicle traffic generation.** In industry, vehicle traffic is usually generated by software tools Chen et al. (2022); Queiroz et al. (2019); Fremont et al. (2019); Jesenski et al. (2019). Some research works Bergamini et al. (2021); Tan et al. (2021); Feng et al. (2023); Rempe et al. (2022) focus on scene-context-conditioned or weakly controlled generation rather than language-conditioned actor-level control. These methods typically condition on road maps, initial states, traffic histories, or sampled goals, and aim to match the distribution of realistic traffic behavior. Recent works on vehicle traffic generation Suo et al. (2021); Zhong et al. (2023b;a); Tan et al. (2023); Lu et al. (2024); Ding et al. (2023) introduce stronger control signals, such as trajectory constraints, language descriptions, scene-control parameters, or retrieval-based priors. Language-conditioned methods such as LCTGen Tan et al. (2023) and CTG++ Zhong et al. (2023a) move toward user-controlled traffic generation, but mainly operate on vehicle trajectories and provide limited support for pedestrians, low-level animation, or fine-grained interaction execution. Similarly, Talk2Traffic Sheng et al. (2025) generates executable simulator scenarios from multimodal instructions, but its output is a Scenic/CARLA-style scenario specification rather than a low-level pedestrian/vehicle animation plan compatible with our pipeline.

However, these works generally do not directly consider physical and kinematic constraints for the traffic and lack sufficient interaction. Several early studies, such as those by Li et al. (2017); Lin et al. (2016); Wang et al. (2018), consider certain potential interaction behaviors in traffic. However, these works are unable to achieve effective control through means such as linguistic commands, and they also lack sufficiently granular modeling of interactions involving pedestrians, such as hailing a taxi or vehicle-pedestrian collisions. Overall, prior traffic-generation methods provide important tools for trajectory or scenario synthesis, but they do not directly provide language-conditioned, participant-level planning together with low-level physics-based pedestrian animation and kinematics-aware vehicle animation. Our LLM-agent planner uses the language-understanding capability of LLMs to improve participant-level control while considering interaction information, and VehAnimator generates the final kinematically compliant vehicle animation.

**Large language models and agents.** Recently, numerous large language models Touvron et al. (2023); Liu et al. (2024); Bai et al. (2023); Achiam et al. (2023) have been proposed and released. Many works have used these models by fine-tuning them Hu et al. (2021); Qiu et al. (2023) or integrating them with relevant tools to build LLM-based agents Liu et al. (2023); Wu et al. (2023). These agents have been applied to a wide range of downstream tasks Hong et al. (2023); Zhou et al. (2023); Li et al. (2024b;a); Leng & Yuan (2023); Shen et al. (2024). In this paper, we explore the application of agents in traffic simulation, using them to interact and perform trajectory and behavior planning within traffic scenarios.

## 3 Method

ChatAni is a language-driven framework for interactive multi-actor animation generation in street scenes, composed of three components: (1) PedAnimator—a unified multi-task controller that produces physically plausible pedestrian animations by integrating interaction signals, trajectory tracking, and upper-body motion. It incorporates interaction task training, task-masking with embeddings for policy unification, Body-masked Adversarial Motion Priors, and hierarchical control to improve animation quality; (2) VehAnimator—a vehicle animation generator that translates high-level raw trajectories into kinematically compliant vehicle motions via a bicycle-model transition and action-to-state control; and (3) a Multi-LLM-Agent Role-Playing Framework—a language-driven planner leveraging LLMs' natural language understanding to construct role-specific interactive agents with inter-agent communication, promoting semantically consistent, language-aligned high-level plans. During execution, traffic participants are initialized as LLM agents that collaboratively establish the scene context through communication. The generated high-level plans are then dispatched to PedAnimator or VehAnimator, which synthesize low-level animations under pedestrian physical constraints or vehicle kinematic constraints, respectively. These outputs are composited into final animations that aim to satisfy the corresponding motion constraints and linguistic specifications. The system overview is shown in Fig. 2. We also provide the corresponding algorithm pseudocode in Appendix S1. In our implementation, PedAnimator is trained in Isaac

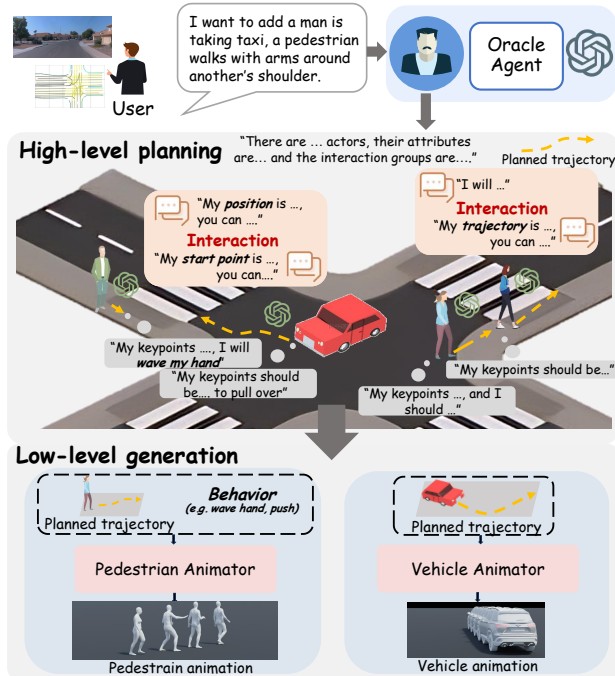

Figure 2: ChatAni employs a multi-agent role-playing scheme for high-level planning. Each traffic participant is modeled as a specific actor that formulates its own animation plan based on natural language instructions and inter-actor temporal interactions. These finalized plans are then dispatched to specialized animators for low-level motion generation.

Gym Makoviychuk et al. (2021) with an SMPL humanoid model Loper et al. (2023); VehAnimator uses the bicycle-model transition described below; and the multi-LLM-agent planner uses GPT-4 Achiam et al. (2023). Rendering follows the ChatSim/Blender pipeline Wei et al. (2024). Additional implementation details, prompts, state definitions, and configuration settings are provided in Appendix S4 and Appendix S6.

## 3.1 PedAnimator

PedAnimator (Fig. 3) is a unified multi-task controller that generates interaction-aware pedestrian animations from interaction targets, planned trajectories, and upper-body motion conditions. Multi-pedestrian interactions constitute critical event elements in traffic scenarios but remain less explored in existing methods. PedAnimator uses a shared observation and action interface across its tasks: task-specific observations encode the planned trajectory, reference motion, or interaction target; a task mask selects the active conditions; and task-specific rewards define the objective for the current episode. This shared interface allows one policy to support trajectory following, motion imitation, and selected interaction tasks. The system also integrates body-masked Adversarial Motion Priors (AMP) and hierarchical action control to improve control fidelity while encouraging human-like animations under the active task constraints.

The control process is defined by the Markov decision process $\mathcal{M}^p = \{\mathcal{S}^p, \mathcal{A}^p, \mathcal{T}^p, \mathcal{R}^p, \gamma^p\}$, whose elements denote the state space, action space, transition function, reward, and discount factor. We use goal-conditioned reinforcement learning (RL), with the transition function implemented by the Isaac Gym physics engine. The following subsections define the policy observations, task masking, hierarchical action representation, and reward design.

**States and Multi-task Unified Training.** Our control policy is designed to handle multiple distinct tasks, requiring specialized processing to achieve a unified policy capable of addressing them. The tasks are categorized into three main groups: trajectory following, single-agent behavior specification, and multi-pedestrian interaction. For single-agent behavior, the LLM generates text descriptions, which are then processed by a Text2Motion model (e.g., MoMask Guo et al. (2024)) to produce upper-body motion for the policy to replicate, specifying single-agent behavior. Multi-agent interaction tasks are trained based on predefined types, with the LLM classifying interaction behaviors, enabling the animator to execute specified interactions. Here, trajectory following means that the pedestrian controller tracks a target root trajectory, which can come from the high-level planner during inference or from reference trajectories during training. Motion imitation means that the controller matches a reference motion signal, such as an upper-body motion clip produced by Text2Motion or sampled from motion data. Thus, PedAnimator performs conditional generation: it generates physically simulated full-body animation while following trajectory and/or motion conditions.

Task-related states consist of trajectory slices $\mathcal{S}^p_{traj}$, motion to be imitated $\mathcal{S}^p_{mo}$, and states of interacting targets $\mathcal{S}^p_{tar}$. $\mathcal{S}^p_{traj}$ includes the K future steps of the planned trajectory, while $\mathcal{S}^p_{mo} = \text{concat}(\hat{j}_{pos}, \hat{j}_{rot}, \hat{j}_{vel}, \hat{j}_{\omega})$ represents joint position $\hat{j}_{pos}$, rotation $\hat{j}_{rot}$, velocity $\hat{j}_{vel}$, and angular velocity $\hat{j}_{\omega}$, with optional joint masking to focus on relevant parts, such as the upper body. $\mathcal{S}^p_{tar} = \text{concat}(r_{pos}, r_{rot}, r_{vel}, r_{\omega}, r_{bbox}, p_{inter}, e_{inter})$ includes the root position $r_{pos}$, root rotation $r_{rot}$, root velocity $r_{vel}$, angular velocity $r_{\omega}$, bounding box vertices $r_{bbox}$, interaction contact position $p_{inter}$, and an interaction embedding $e_{inter}$ as a one-hot vector specifying the interaction characteristic. The target-state format is shared across behaviors, while the interaction embedding and task-specific reward distinguish their objectives. Interaction tasks use the shared target-state representation together with task-specific rewards. The contact position $p_{inter}$ specifies where contact should occur, the interaction embedding $e_{inter}$ identifies the behavior type, and the corresponding reward specifies the objective, such as toppling a target, reaching a contact point, or maintaining contact while following a trajectory. Combining these interaction conditions with trajectory and motion states allows the unified policy to execute the selected interaction behavior. In addition to task-related states, the final observation includes humanoid proprioception Wang et al. (2024) $\mathcal{S}^p_{prop}$, capturing the internal states of the humanoid. The final state is represented as $\mathcal{S}^p = \text{concat}(\mathcal{S}^p_{traj}, \mathcal{S}^p_{mo}, \mathcal{S}^p_{tar}, \mathcal{S}^p_{prop})$. These state vectors are policy observations rather than rendered pixels. At each simulation step, the policy observes $\mathcal{S}^p$, outputs latent or PD-control actions, and the physics simulator updates the SMPL joint poses over time, producing the final animated motion sequence. Because PedAnimator uses one unified policy for several task types, not

every control signal is relevant in every episode. For example, a pure trajectory-following task uses $\mathcal{S}^p_{traj}$ but does not require motion-imitation or interaction-target states, while an interaction task may activate both target states and trajectory conditions. We therefore introduce a task masking mechanism that activates only the state and reward terms required by the current task and masks the remaining task-specific inputs. During training, tasks for each episode are sampled, and the corresponding task-related states are unmasked while others are masked. During inference, relevant states are activated while others are masked, supporting task-specific execution. This approach helps the unified policy handle different tasks without confusion. After sufficient training, the policy performs at the level of individually trained policies while being capable of executing multiple non-conflicting tasks in a single run.

Fig. 3 summarizes how the PedAnimator components interact as a closed-loop controller. The task-specific states provide trajectory, reference-motion, and interaction-target conditions; the task mask selects the active conditions for the current episode; the policy maps the resulting observation to an action representation; the hierarchical action module converts this representation into joint-level control targets; and the physics simulator updates the humanoid state. During training, task rewards specify what the character should accomplish, while AMP rewards regularize how the motion should look.

**Action hierarchical control.** Pedestrian action spaces are typically modeled using a proportional-derivative (PD) controller at each degree of freedom (DoF), but directly outputting unconstrained joint-level PD targets gives the policy a high-dimensional action space and can produce uncoordinated or locally unnatural joint commands. To reduce this issue, we adopt the hierarchical action control space from PULSE Luo et al. (2024). PULSE is used here as an existing module rather than a contribution of this work. The prior refers to a data-driven latent action space learned from human motion data, where latent actions decode to coordinated, human-like joint-control patterns. The policy network first outputs a latent feature $\mathbf{f_{action}}$, which is then mapped by the pretrained PULSE decoder into joint-level control signals. This biases the action search space toward motions observed in human data, stabilizes RL training, and reduces locally unnatural poses, while task rewards still determine the following, imitation, or interaction objective.

**Reward design.** The reward consists of two main components with different roles. The task-related reward $R_{task}$ specifies what the character should accomplish, such as following a trajectory, imitating a reference motion, or reaching an interaction contact target. The discrimination reward $R_{disc}$ implements AMP Peng et al. (2021), which uses a discriminator to regularize how the motion should look by encouraging generated outputs to align with movement patterns observed in human-recorded data clips. This separation lets the controller satisfy task conditions without drifting too far from human-like motion statistics. Detailed task reward designs are provided in Appendix S4.2. Similar to Luo et al. (2023), we use early termination when excessive root-distance or joint-distance errors occur, and we include a fail-state recovery task to improve robustness under perturbations. The recovery task also plays an important role in low-level interactions between pedestrians and vehicles.

**AMP with body mask and warm-up training.** We employ Proximal Policy Optimization (PPO) Schulman et al. (2017) for overall training optimization. However, directly optimizing for interaction tasks presents certain challenges: without AMP, the lack of a discrimination reward makes it difficult to achieve human-like animation results. Conversely, proper reference data clips for interaction tasks are usually complex and not easily available. Here, following and imitation refer to the trajectory-conditioned and reference-motion-conditioned tasks introduced above. Their reference clips provide useful locomotion or upper-body motion priors, but they do not necessarily contain the contact patterns needed for interaction tasks such as pushing or patting. Directly using such clips as AMP references for interaction training can therefore encourage motions that look plausible in isolation but fail to satisfy the interaction objective.

We address these challenges through two strategies: (i) A two-phase approach: initial AMP-free training establishes task-completion foundations for subsequent AMP-enhanced optimization that preserves functionality while improving motion plausibility; (ii) AMP body masking that excludes interaction-focused joints (e.g., arms) during discriminator calculations, enabling AMP-driven natural motion for non-interactive body parts while maintaining task-specific joint flexibility. These techniques collectively encourage visually plausible animations with effective interaction task performance.

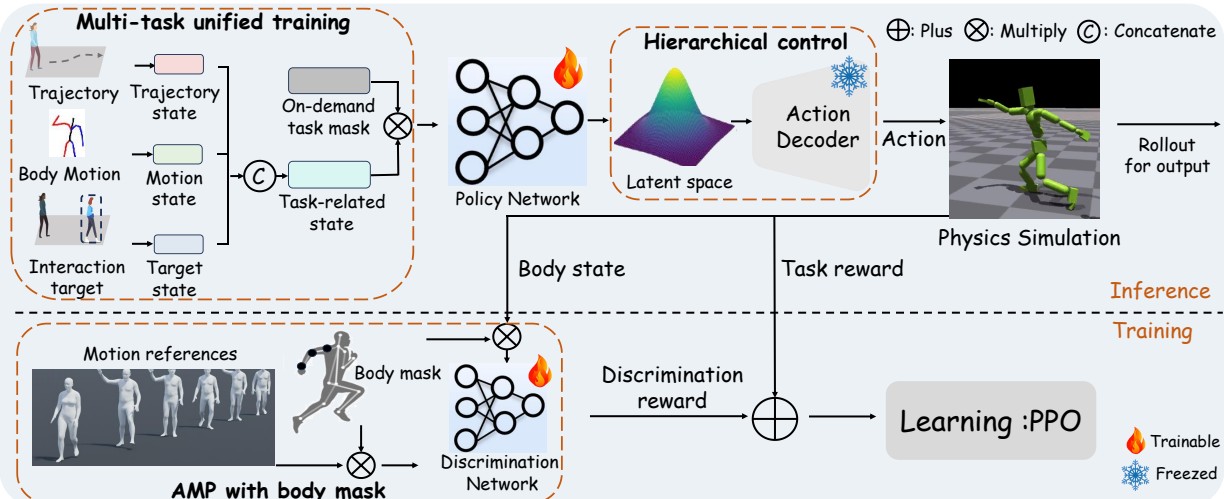

Figure 3: Pedestrian animator (PedAnimator) framework. With multi-task unified training, PedAnimator supports trajectory following, motion imitation, and selected interaction tasks. The figure shows the flow from task states and the on-demand task mask to the policy, hierarchical action decoder, physics simulation, and task/discrimination reward feedback. The dashed horizontal divider separates the inference and training paths. The legend marks trainable modules with the fire symbol and frozen modules with the snowflake symbol. Hierarchical control and body-masked AMP provide action and motion priors. Tab. 3, Tab. 4, Tab. S2, and the supplementary video report the corresponding quantitative and perceptual evaluations.

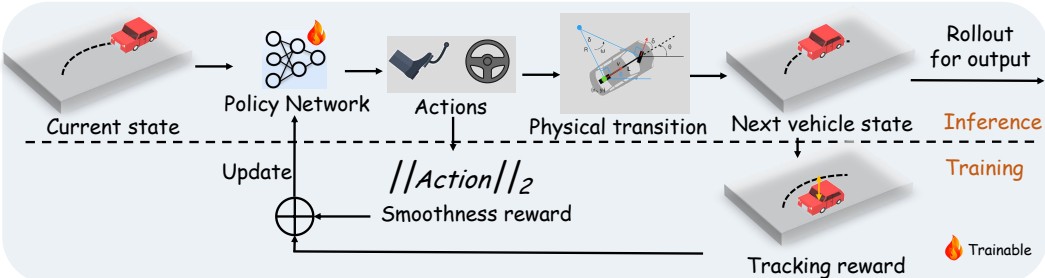

Figure 4: Vehicle animator (VehAnimator) framework. VehAnimator adopts goal-conditioned RL with a bicycle-model transition and generates kinematically compliant vehicle animation from planned trajectories. The dashed horizontal divider separates the inference and training paths, and the legend marks the trainable policy network with the fire symbol. Tab. 5, Tab. S1, Fig. S5, and the supplementary video report its trajectory-tracking errors, obstacle-avoidance examples, and user-preference results.

### 3.2 VehAnimator

VehAnimator, as shown in Fig. 4, translates raw planned vehicle trajectories into kinematically compliant animations. Raw trajectories do not necessarily satisfy a vehicle transition model and can produce artifacts such as excessive lateral motion, tail swing, or drifting when applied directly. VehAnimator instead predicts steering and acceleration controls and applies the bicycle-model transition to update the vehicle state. This process enforces the stated kinematic transition while the history-aware observation design improves tracking accuracy and temporal consistency. The control process is also modeled as a Markov decision process defined by $\mathcal{M}^v = \{\mathcal{S}^v, \mathcal{A}^v, \mathcal{T}^v, \mathcal{R}^v, \gamma^v\}$, with goal-conditioned RL for training.

**History-aware states.** The VehAnimator incorporates historical information into its states, alongside its current state, to improve the temporal consistency of the policy. The vehicle states consist of the planned

trajectory segment $\hat{\mathbf{P}}^v$, temporal velocity $\mathbf{V}^v$, and dynamic parameters $\Theta^v$. $\mathbf{P}^v$ is a slice of the planned trajectory in the vehicle's coordinate system over the current and adjacent $\tau^v$ frames. $\mathbf{V}^v$ represents the vehicle's centroid velocity over $\tau^v$ frames. $\Theta^v$ includes inherent vehicle parameters such as $L$ (vehicle length), $W$ (vehicle width), $l_f$ (front overhang), and $l_r$ (rear overhang). These parameters provide prior information that influences the dynamic transition process.

**Vehicle actions.** To model vehicle actions and maintain temporal consistency, the vehicle action space $\mathcal{A}^v$ is defined by the delta steering angle $\Delta\delta$ and scalar acceleration $a$, which are the two most direct controls affecting vehicle movement in actual driving.

**Vehicle transition function.** We employ the bicycle model Rajamani (2012) to model the vehicle dynamic transition process. Let $x$ and $y$ denote the vehicle's coordinates, $v$ represent the scalar velocity, and $\phi$ indicate the vehicle's orientation. Utilizing the inherent parameters from the states, the vehicle's state transition process can be expressed as:

$$\dot{x} = v\cos(\phi + \beta), \ \dot{y} = v\sin(\phi + \beta), \ \beta = \arctan\left(\frac{l_r}{l_f + l_r}\tan(\delta)\right), \ \dot{\phi} = \frac{v}{l_f + l_r}\cos(\beta)\tan(\delta).$$

where $\beta$ denotes the tire slip angle and $\dot{}$ denotes a time derivative. The bicycle model simulates dynamic vehicle-state changes induced by vehicle actions.

**Reward and training.** The reward focuses on two aspects: following the planned trajectory and maintaining smooth actions. Therefore, the reward consists of two components: $R_{pos}^v = -||\hat{p}_t - p_t||_2$ for following the planned trajectory and $R_{act}^v = -(c_\delta||\Delta\delta||_2 + c_a||a||_2)$ for smoothness, where $c_\delta$ and $c_a$ balance the two terms with different units. Training uses TD3 Fujimoto et al. (2018) to maximize the accumulated discounted reward. Note that: (i) actions can be further smoothed with temporal filtering: $\mathcal{A}_t^v = \alpha\mathcal{A}_{t-1}^v + (1-\alpha)\mathcal{A}_{policy}^v$, where $\mathcal{A}_t$ is the action taken at timestep $t$ and $\mathcal{A}_{policy}$ is the action directly output by the policy network; (ii) obstacles can be considered by concatenating their positions and radii in the state and adding $R_{obs}^v = \frac{\epsilon}{L_o}$ to the reward if $L_o < D$, where $L_o$ is the distance to the obstacle, $\epsilon < 0$ is a penalty coefficient, and $D$ is a threshold.

### 3.3 Multi-LLM-agent role-playing system design

PedAnimator and VehAnimator facilitate low-level control for pedestrian and vehicle animations. For language-based holistic scene control, we developed a multi-LLM-agent role-playing framework that generates high-level planning for all participants from user input, offering control signals for low-level animation. The framework includes agents with LLM-based reasoning and tool interfaces, divided into two types: (1) Oracle Agent processes user language inputs to create global scene context and scheduling instructions; (2) Actor Agents represent individual traffic participants, initialized with Oracle Agent information, and engage in inter-agent communication to collaboratively develop high-level plans. This design utilizes the language understanding and domain knowledge of LLM to interpret user input. Actor Agents dynamically adjust planning based on participant types (e.g., pedestrians, vehicles), meeting diverse behavioral needs through specific tool functions while supporting scene-level coordination. See Fig. 2 for details.

#### 3.3.1 Oracle Agent

The Oracle Agent decomposes complex user instructions into participant-specific actions, supporting structured interpretation and schedule generation. Its LLM component is prompted with role definitions, requirements, and few-shot examples, and produces JSON-formatted structured outputs whose selected text fields contain natural-language behavior descriptions. Deterministic follow-up functions convert these outputs into executable predefined data structures. Oracle Agent enables the system to process composite abstract instructions while improving operational clarity and granularity. See more details in Appendix S6.

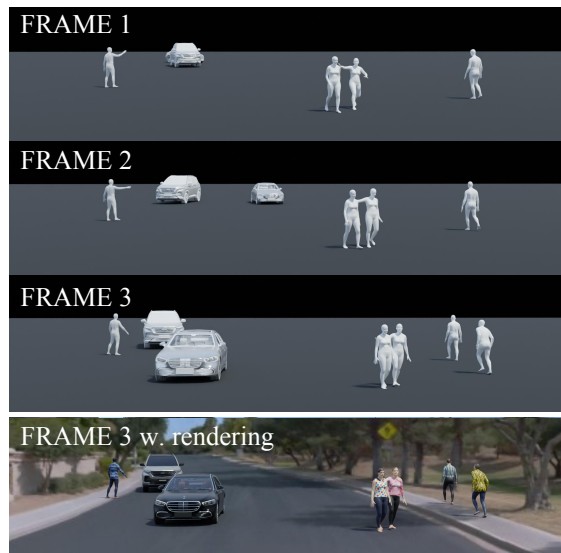
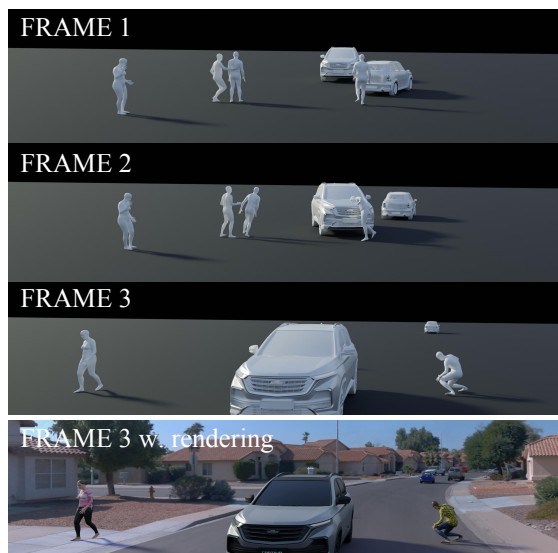

**Command:** *A person is taking a taxi at the left roadside, and a vehicle overtakes the taxi. Two persons are walking together with one's arm around another's shoulder. A person is chasing another person along the roadside.*

**Command:** *A person pushes another person, and another person is making a phone call, then walks along the roadside. A vehicle turns right at the intersection and hits a man, and a hurried vehicle changes lane.*

Figure 5: ChatAni examples under abstract, multi-faceted instructions. By bridging high-level semantic reasoning with low-level physical synthesis, the system generates scenarios with diverse interactions, including pedestrian-vehicle, multi-pedestrian, and multi-vehicle encounters. The framework supports both semantic-level coordination (e.g., intent-based yielding) and low-level physical interactions (e.g., collision avoidance and gait generation). To evaluate the visualization quality through quantitative experiments, Tab. 6 shows the quantitative command-following results of ChatAni. Tab. 3, Tab. 4 and Tab. S2 report the accuracy and user preference results of PedAnimator. The tracking accuracy and user preference results are shown in Tab. 5 and Tab. S1.

### 3.3.2 Actor Agent

Each Actor Agent corresponds to a participant in the scene, initialized with information from Oracle Agent, including agent type, trajectory, and behavioral descriptions. These agents communicate according to the Oracle Agent schedule to collect interactive information and formulate final high-level planning. The planning process incorporates map data, where the scene map is represented as a graph $\mathcal{G} = (\mathcal{N}, \mathcal{E})$. $\mathcal{N}$ denotes lane sections, each containing a point set and metadata such as orientation and driving type (e.g., straight, turn, lane/boundary). Edges $\mathcal{E}$ encode relationships between lane sections (e.g., adjacency, connectivity). The planning adopts a keypoints-based methodology, synthesizing actor interactions to refine the final trajectory.

**Keypoints-based trajectory planning.** Trajectories are defined by keypoints interpolated into continuous paths. The Actor Agent's LLM component determines the required keypoint count and their generation logic based on initialization information. Keypoints originate from two sources: (1) Map retrieval: extracted from the map $\mathcal{G}$ by analyzing attributes (e.g., lane orientation, driving type) and topological relationships (via edges $\mathcal{E}$) between nodes $\mathcal{N}$; (2) Interaction derivation: determined through inter-agent communication, where dependencies on other participants' states influence keypoint placement. This hybrid approach ensures trajectory coherence with both environmental constraints and multi-agent interactions.

**Inter-agent communication.** Communication allows agents to acquire information from others to generate interaction-dependent keypoints. Predefined interaction groups, through the Oracle Agent, guide the process. The LLM identifies the required parameters (e.g., start/end points, trajectories) from interacting agents. Agents exchange requested data, which are processed by tools to generate keypoints. This mechanism supports interactive trajectory generation. The communication is limited to a maximum of 5 rounds to prevent infinite loops.

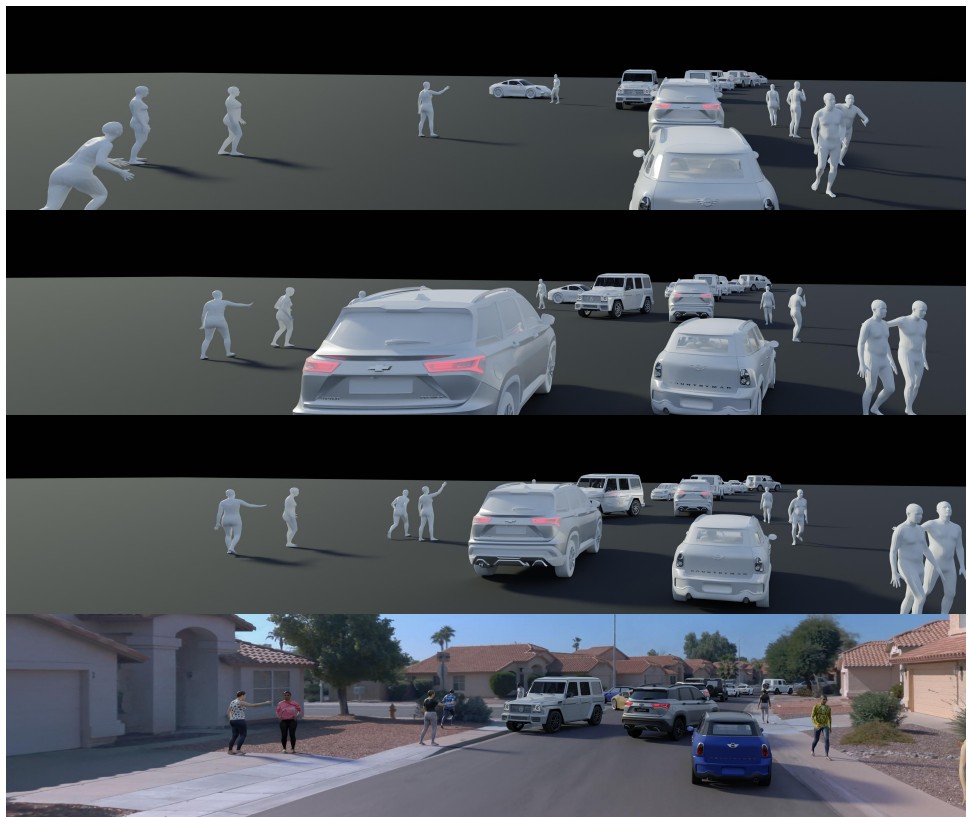

**Command:** *Make the scene crowded like a traffic jam. On the left, one person has slipped, and another is about to help.*

Figure 6: System results under an abstract command with large-scale output. This example illustrates ChatAni's ability to handle abstract semantics and manage a larger scenario in the evaluated setting. Quantitative results for controllability, accuracy, and user preference are shown in Tab. 3, Tab. 4, Tab. S2, Tab. 5, Tab. S1, and Tab. 6. The supplementary video includes the corresponding animation.

Pedestrian animations of **pushing** task

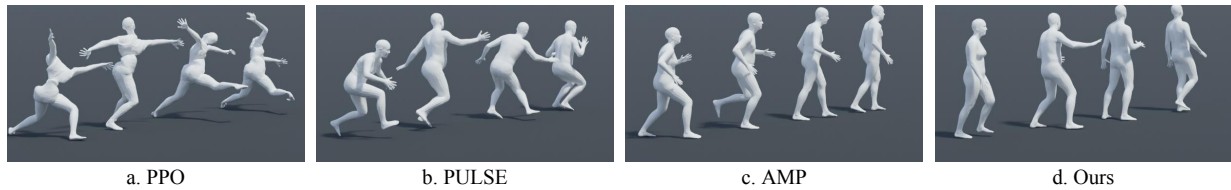

| a. PPO | b. PULSE | c. AMP | d. Ours |

Figure 7: Compared to baseline methods, PedAnimator executes the complex pushing task with improved visual quality in this example. While existing approaches are often limited to simple task completion or mere kinematic imitation—frequently resulting in unnatural postures or jittery motions—our method completes the task with more natural motion in the shown comparison. To quantitatively validate the quality of the associated visualizations, the results in Tab. 4 show the execution success rate of our method, and Tab. S2 shows the animation quality by user preference. The related comparison video can also be found in the supplementary video.

Bézier curve interpolation aggregates keypoints to generate complete planning trajectories, with three operational notes: (1) interpolated trajectories serve as physics-agnostic plans for animators; (2) static actors require two orientation-defining keypoints; (3) pedestrian agents have LLM-inferred behavioral instructions

(e.g., "wave hand while walking") from initialization data—derived from explicit commands/implicit semantic context—formatted as executable text directives.

## 4 Experiments

### 4.1 Implementation Details and Dataset

PedAnimator is trained in Isaac Gym Makoviychuk et al. (2021) as the physics simulator, using AMASS Mahmood et al. (2019) as reference motion data and an SMPL Loper et al. (2023) humanoid as the simulated body. The LLM planner uses the GPT-4 API Achiam et al. (2023). Final rendering follows the Chat-Sim/Blender pipeline Wei et al. (2024). PedAnimator and VehAnimator are trained independently with separate environments and model parameters. Detailed settings, configurations, additional experiments, video results, and selected code components are provided in the appendix and supplementary materials.

### 4.2 System Results

In this section, we evaluate the generation capabilities of the proposed ChatAni framework. The evaluation is structured around three aspects. First, we show representative full-system results for interactive, physically plausible, and controllable traffic scenarios. Second, we examine the framework's scalability and its capacity to interpret abstract semantics in large-scale, crowded traffic environments. Third, we add an end-to-end human evaluation across diverse prompts, scenes, and interaction types. Together, these results provide evidence for ChatAni's ability to handle complex and diverse traffic behaviors in the evaluated settings.

#### 4.2.1 Interactive, plausible and controllable

We showcase representative keyframes from two simulated scenarios in Figure 5, each containing various traffic participants generated according to commands. The results illustrate three capabilities in the evaluated scenarios: (i) interaction is depicted across pedestrian-vehicle (crashing, taxi hailing, collision avoidance, speed adaptation), vehicle-vehicle (lane switching, overtaking), and pedestrian-pedestrian (pushing, chasing, walking with arm around shoulder) cases, which enrich the scene animations and cover more diverse event types; (ii) PedAnimator generates physically plausible pedestrian motion with physical feedback, including recovery from falling, while VehAnimator generates kinematically compliant vehicle motion; (iii) command-following control is supported by the multi-LLM-agent framework through decomposition of complex descriptions containing abstract semantics into executable instructions. Combining these features, the generated animations can represent selected dangerous corner cases that are otherwise difficult to obtain.

#### 4.2.2 Scalable and abstract understanding

We illustrate ChatAni's ability to understand and decompose abstract semantics in 6, while also supporting diverse interaction behaviors. At the same time, this result shows ChatAni's scalability in this large-scale scene traffic generation example, as it handles a large number of traffic participants simultaneously present in the scene, constructing crowded and complex traffic scenarios through the interactions of numerous people, vehicles, and their behaviors.

#### 4.2.3 End-to-end full-system evaluation

To evaluate ChatAni as a complete system beyond selected qualitative examples, we conduct an end-to-end human evaluation across diverse prompts, scenes, and interaction types. We generate 120 full-system results across six categories, with 20 generated scenes per category. The categories are defined as follows: *single-agent* prompts specify the behavior of one participant; *vehicle-vehicle* prompts involve interactions such as overtaking, lane switching, and speed adaptation; *pedestrian-vehicle* prompts involve crossing, hailing, collision avoidance, or collision-related events; *pedestrian-pedestrian* prompts involve interactions such as pushing, chasing, patting, or walking together; *compound* prompts combine multiple participants and interaction requirements; and *crowd/abstract* prompts use higher-level descriptions of crowded or complex traffic situations.

Table 2: End-to-end full-system human evaluation of ChatAni across diverse prompts, scenes, and interaction types. Each category contains 20 generated scenes, and each generation is evaluated by five participants. We report the percentage of positive ratings with Wilson 95% confidence intervals.

| Category | Actor Cov. | Cmd. Match | Interact. Succ. | Map Valid. | Scene Succ. |
|---|---|---|---|---|---|
| Single-agent | 96.0 [90.2, 98.4] | 94.0 [87.5, 97.2] | 92.0 [85.0, 95.9] | 93.0 [86.3, 96.6] | 89.0 [81.4, 93.7] |
| Vehicle-vehicle | 92.0 [85.0, 95.9] | 87.0 [79.0, 92.2] | 85.0 [76.7, 90.7] | 89.0 [81.4, 93.7] | 81.0 [72.2, 87.5] |
| Pedestrian-vehicle | 90.0 [82.6, 94.5] | 84.0 [75.6, 89.9] | 81.0 [72.2, 87.5] | 86.0 [77.9, 91.5] | 77.0 [67.8, 84.2] |
| Pedestrian-pedestrian | 88.0 [80.2, 93.0] | 83.0 [74.5, 89.1] | 78.0 [68.9, 85.0] | 84.0 [75.6, 89.9] | 73.0 [63.6, 80.7] |
| Compound | 84.0 [75.6, 89.9] | 77.0 [67.8, 84.2] | 72.0 [62.5, 79.9] | 79.0 [70.0, 85.8] | 65.0 [55.3, 73.6] |
| Crowd/abstract | 81.0 [72.2, 87.5] | 71.0 [61.5, 79.0] | 65.0 [55.3, 73.6] | 74.0 [64.6, 81.6] | 58.0 [48.2, 67.2] |
| Overall | 88.5 [85.7, 90.8] | 82.7 [79.4, 85.5] | 78.8 [75.4, 81.9] | 84.2 [81.0, 86.9] | 73.8 [70.2, 77.2] |

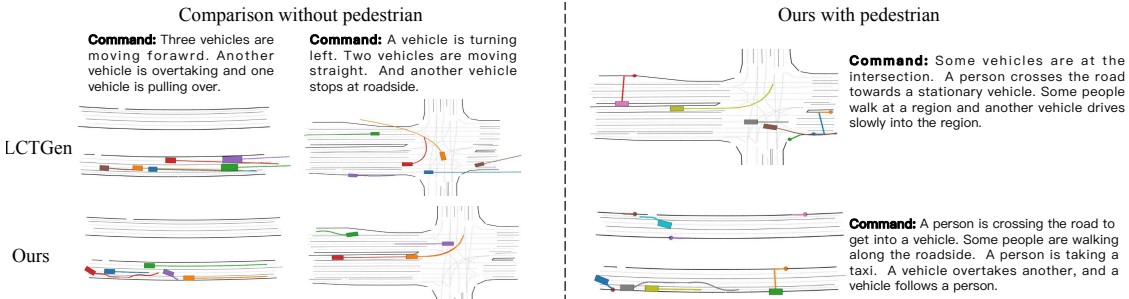

Figure 8: High-level planning results comparison under vehicle-only command, and our planning results with pedestrians involved. The boxes indicate vehicles and the circles indicate pedestrians. Our method produces plans with better command-following and road-boundary compliance in the evaluated benchmark. Section 4.3.3 states more quantitative results, and Tab. 6 shows the control accuracy and command following comparison of our method.

We recruit 20 participants for the evaluation. Each participant evaluates 30 randomly assigned prompt-result pairs, resulting in five evaluations from different participants for each generated scene and 600 prompt-result evaluations in total. Because each evaluation contains five binary questions, the study collects 3,000 binary judgments across the five criteria. For each prompt-result pair, participants provide judgments on actor coverage, command matching, interaction success, map validity, and overall scene success. Tab. 2 reports the percentage of positive judgments for each criterion with Wilson 95% confidence intervals. Overall, ChatAni obtains 88.5% actor coverage, 82.7% command matching, 78.8% interaction success, 84.2% map validity, and 73.8% scene success across all evaluated generations. Performance is strongest on single-agent prompts and decreases on compound and crowd/abstract prompts, reflecting the increased difficulty of these scenarios.

Representative failure cases and their analysis are provided in Appendix S10.

## 4.3 Component Results

In this section, we present a detailed evaluation of the individual modules within the proposed framework. The assessment is divided into three parts, corresponding to the core components: PedAnimator, VehAnimator, and the Multi-LLM-Agent planning module. For each component, we conduct quantitative comparisons and user studies against established baselines to measure specific performance metrics, such as motion imitation accuracy, trajectory following error, and command compliance. These component-level evaluations assess the behavior and accuracy of each subsystem under the evaluated settings.

Because ChatAni combines language-conditioned planning, map-constrained multi-actor traffic control, physics-based pedestrian animation, kinematics-aware vehicle animation, and pedestrian/vehicle interactions, we are not aware of an existing method that is directly compatible with the complete setting. We therefore adapt the closest baselines to our scenarios and shared interfaces while retaining their core models,

| Methods | FID↓ | Div.↑ | l-FID↓ | l-Div.↑ | $E_{mpjpe}\downarrow$ | $E_f\downarrow$ | UP↑ |
|---|---|---|---|---|---|---|---|
| Pacer | $7.25 \pm 0.46$ | $1.24 \pm 0.09$ | $7.93 \pm 0.52$ | $1.05 \pm 0.07$ | \ | $0.122 \pm 0.009$ | 0.137 [0.125, 0.149] |
| Pacer+ | $6.62 \pm 0.31$ | $1.58 \pm 0.12$ | $7.76 \pm 0.47$ | $1.28 \pm 0.10$ | $82.33 \pm 3.84$ | $0.128 \pm 0.011$ | 0.218 [0.204, 0.232] |
| UniPhys Wu et al. (2025) | $6.34 \pm 0.33$ | $1.69 \pm 0.13$ | $7.21 \pm 0.39$ | $1.42 \pm 0.10$ | $81.06 \pm 3.45$ | $0.126 \pm 0.010$ | 0.296 [0.281, 0.312] |
| Ours | $\mathbf{6.21 \pm 0.28}$ | $\mathbf{1.76 \pm 0.11}$ | $\mathbf{7.07 \pm 0.36}$ | $\mathbf{1.49 \pm 0.09}$ | $\mathbf{79.82 \pm 3.17}$ | $0.124 \pm 0.008$ | **0.349 [0.333, 0.366]** |

Table 3: Quantitative Evaluation of Animation Quality and Accuracy for Trajectory Following and Motion Imitation. Continuous metrics are reported as mean ± standard deviation over three independent runs; user-preference scores are reported with Wilson 95% confidence intervals. PedAnimator achieves lower FID, lower l-FID, lower MPJPE, and higher diversity/user-preference scores than the compared baselines in this evaluation, while obtaining a comparable following error.

| Methods | Unified policy | Interaction 1 | Interaction 2 | Interaction 3 |
|---|---|---|---|---|
| PPO | × | 0.971 [0.959, 0.980] | 0.934 [0.917, 0.948] | 0.914 [0.895, 0.930] |
| AMP | × | 0.234 [0.209, 0.261] | 0.179 [0.156, 0.204] | 0.108 [0.090, 0.129] |
| PULSE | × | 0.975 [0.963, 0.983] | 0.942 [0.926, 0.955] | 0.925 [0.907, 0.940] |
| Ours | ✓ | **0.982 [0.972, 0.989]** | **0.977 [0.966, 0.985]** | **0.971 [0.959, 0.980]** |

Table 4: Success Rate Evaluation of Various Interaction Tasks. Each cell is evaluated over 1,000 episodes and reports the success rate with a Wilson 95% confidence interval. The results show that PedAnimator can execute the evaluated interaction tasks with a unified policy while maintaining high task success rates. User preference and visual comparisons are used to assess the perceived quality of the generated motions.

and restrict each comparison to the metrics that the corresponding baseline can support. For pedestrian animation, we compare with Pacer Rempe et al. (2023), Pacer+ Wang et al. (2024), and UniPhys Wu et al. (2025) for trajectory-following and motion-imitation metrics, while UniPhys is a general physics-based character-control method and does not directly support traffic-map interactions or pedestrian-vehicle interaction tasks. For high-level planning, we compare with LCTGen Tan et al. (2023), ChatSim Wei et al. (2024), and Talk2Traffic Sheng et al. (2025), while Talk2Traffic produces Scenic/CARLA-style scenarios rather than low-level animation plans for our Waymo-map-based pipeline. For vehicle animation, we compare with classical tracking and learning-based vehicle-control baselines.

### 4.3.1 PedAnimator

We evaluate PedAnimator on trajectory following and motion imitation tasks with our test motion dataset (about 2000 samples), as shown in Table 3. The quality and diversity of the generated results are measured by Fréchet Inception Distance (FID) Heusel et al. (2017) and diversity metric (Div.) Wang et al. (2024) at normal speed, and by l-FID and l-Div. at low speed. Imitation accuracy is measured by Mean Per-Joint Position Error (MPJPE), while following accuracy is evaluated using following error ($E_f$). User preference (UP) is assessed by 100 participants, each evaluating 33 randomly ordered animation comparisons with anonymized method identities. With hierarchical control and the associated training strategy, PedAnimator shows competitive performance in both following and imitation tasks under these metrics. More user-study details are provided in Appendix S9.

We evaluate interaction task performance across methods, comparing PedAnimator's unified policy with task-specific training used by the baselines (Table 4, Fig. 7). In the three evaluated interaction tasks – pushing, patting, and arm-around-shoulder walking – AMP Peng et al. (2021) produces reference-like motions but lacks task-specific interaction data, yielding superficial mimicry without task completion. PPO Schulman et al. (2017) and PULSE Luo et al. (2024) achieve relatively high success rates, but their animations exhibit unnatural movements in our visual comparisons. PedAnimator's body-masked AMP and hierarchical control support both high task success and more human-like motion quality through integrated physical constraints and style preservation. Additional results, including ablations and more visualizations, are provided in Appendix S4.6 and the supplementary video.

| Methods/Speed | 0 | 5 | 10 | 20 |
|---|---|---|---|---|
| PP | $(0.129\pm0.011)/(0.103\pm0.007)$ | $(0.143\pm0.009)/(0.125\pm0.012)$ | $(0.162\pm0.015)/(0.142\pm0.010)$ | $(0.231\pm0.026)/(0.208\pm0.018)$ |
| Xu et al. | $(0.075\pm0.006)/(0.054\pm0.004)$ | $(0.084\pm0.007)/(0.066\pm0.006)$ | $(0.095\pm0.009)/(0.077\pm0.005)$ | $(0.138\pm0.012)/(0.114\pm0.010)$ |
| Ours | $\mathbf{(0.059\pm0.004)/(0.037\pm0.003)}$ | $\mathbf{(0.062\pm0.005)/(0.041\pm0.002)}$ | $\mathbf{(0.077\pm0.006)/(0.054\pm0.005)}$ | $\mathbf{(0.106\pm0.009)/(0.088\pm0.007)}$ |

Table 5: Quantitative Evaluation of Position and Velocity Errors. Each cell reports position/velocity error as mean $\pm$ standard deviation over three independent runs. The results show that VehAnimator achieves the lowest position and velocity errors among the compared baselines across the evaluated speed settings, indicating improved trajectory adherence and velocity tracking in this benchmark.

| Methods | Language command category | | | Within road | User preference |
|---|---|---|---|---|---|
| | single | interaction | compound | | |
| LCTGen | 91.90±0.47 | 20.70±0.70 | 64.10±0.82 | 59.90±0.84 | 15.16±0.62 |
| ChatSim | 84.60±0.62 | 5.79±0.40 | 77.10±0.72 | 86.10±0.59 | 5.53±0.39 |
| Talk2Traffic Sheng et al. (2025) | 92.82±0.44 | 72.05±0.77 | 80.53±0.68 | 85.05±0.61 | 28.36±0.77 |
| Ours | **93.70±0.42** | **87.60±0.57** | **88.40±0.55** | **92.60±0.45** | **50.95±0.86** |

Table 6: High-level Planning Evaluation. We evaluate ChatAni based on command matching rate, within-road rate, and human preferences. The values shown in the table are percentages with Wilson 95% confidence-interval half-widths, computed from $5 \times 26 \times 100$ responses for each cell. The results indicate that ChatAni produces more command-aligned and contextually rich high-level plans than the compared methods in this benchmark. Furthermore, our framework obtains higher within-road rate, command matching rate, and subjective user preference scores among the evaluated methods.

### 4.3.2 VehAnimator

As shown in Table 5, we evaluate the error between VehAnimator-generated animations and reference trajectories at different initial velocities to measure animation accuracy, comparing with Pure Pursuit Craig Coulter (1992) and Xu et al. Xu & Yu (2023). VehAnimator achieves lower errors than the compared methods in these cases. Additional experiments, including ablation studies and more visualizations, are provided in Appendix S4.

### 4.3.3 Multi-LLM-Agent Planning

We benchmark LLM-agent trajectory planning against language-based traffic generation methods (LCT-Gen Tan et al. (2023), ChatSim Wei et al. (2024), and Talk2Traffic Sheng et al. (2025)). Because LCTGen and ChatSim do not support our pedestrian-involved animation format and Talk2Traffic outputs Scenic/CARLA-style scenarios rather than Waymo-map-based animation plans, the quantitative comparison focuses on the shared vehicle-only planning setting. Three instruction categories are evaluated: single vehicle, interaction, and composite. Using 5 maps with 26 samples per category, all 100 participants evaluate every sample for description matching, road boundary compliance, and preference, resulting in $5 \times 26 \times 100$ responses for each table cell, as shown in Table 6. ChatAni obtains higher command-aligned planning and user preference scores among the evaluated methods. Visual comparisons in Fig. 8 (including pedestrian-involved cases) illustrate ChatAni's handling of complex requirements, contrasting with LCTGen's limitations in interaction control and complex command execution. More details on participant assignment, randomization, anonymization, and preference calculation are provided in Appendix S9.

### 4.4 Applications to Autonomous Driving

In this section, we investigate the potential utility of the proposed framework by applying its generated data to downstream tasks. The evaluation focuses on two primary applications of data augmentation: trajectory and human motion prediction, and Vision-Language Model (VLM)-based driving scene understanding. These experiments assess whether the synthesized data provides measurable gains under controlled downstream settings, rather than establishing a general safety improvement claim.

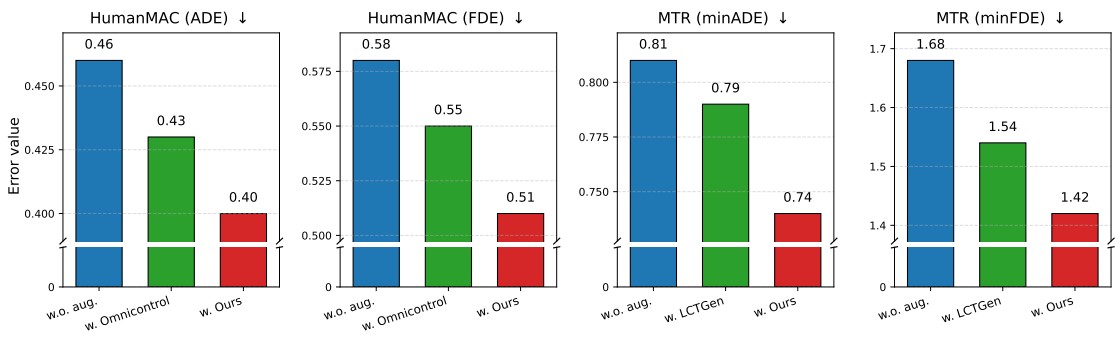

Figure 9: Prediction augmentation for traffic prediction and human keypoints prediction. In the evaluated tasks and splits, ChatAni-based augmentation improves prediction performance and obtains larger gains than the compared augmentation baselines under the same augmentation budgets. The y-axes use explicitly marked breaks for readability, and exact values are annotated on the bars.

### 4.4.1 Augmentation for prediction task

We apply ChatAni for data augmentation in two prediction tasks using seed 1111. For traffic prediction (minADE/minFDE), using MTR Shi et al. (2022) on the Waymo Open Dataset Sun et al. (2020), we train on 243k scenes sampled from the official training split and evaluate on the official validation split. We augment the training set with 20% additional data (48k scenes) generated by LCTGen Tan et al. (2023) or ChatAni. The synthetic samples are generated only from training-split scenes, and no validation scenes are used for augmentation. Both augmentation methods use the same data budget and the same MTR training configuration. For human motion prediction (ADE/FDE), using HumanMAC Chen et al. (2023) on Human3.6M Ionescu et al. (2013), we follow the HumanMAC split and preprocessing protocol, augmenting only the training split with 20% additional samples (2k) from Omnicontrol Xie et al. (2023) or PedAnimator. Both methods improve performance in this single-seed evaluation, while PedAnimator's physically compliant outputs produce larger gains under this setup. Additional protocol details are provided in Appendix S7.

### 4.4.2 Augmentation for VLM-based driving scene understanding

To evaluate the effect of hazardous scenario generation on driving-scene understanding, we conduct a controlled DriveLM Sima et al. (2024) evaluation with seed 1111. We select 30 scenes from the Waymo Open Dataset Sun et al. (2020) training split as held-out evaluation scenes for this downstream test and run 10 trials per scene under three conditions: (1) original scenes, (2) ChatAni-edited scenes with intentionally heightened danger, and (3) edited scenes after fine-tuning with 3000 frames of ChatAni-augmented hazardous data. The 30 evaluation scenes are excluded from the fine-tuning data, and no scene/frame overlap

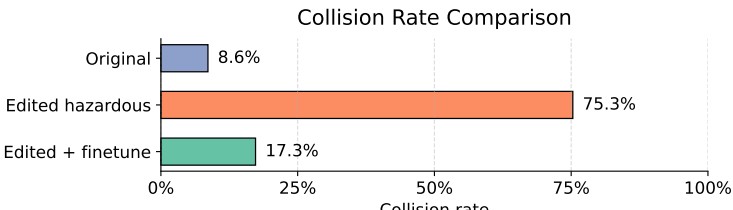

Figure 10: Collision rates of DriveLM Sima et al. (2024) under different scenarios. Bar labels and the x-axis are both shown as percentages. In this evaluated setting, ChatAni-augmented fine-tuning reduces the collision rate on the generated hazardous scenes.

is used between the augmented fine-tuning set and this evaluation set. A collision is counted when the ego vehicle collides with any generated or existing pedestrian or vehicle in the scene. Across 300 trials per condition, the collision rates are 8.6% [5.9, 12.3] for the original scenes, 75.3% [70.1, 79.8] for the ChatAni-edited scenes, and 17.3% [13.4, 22.0] after augmented fine-tuning, where brackets denote aggregate Wilson

95% confidence intervals. The ChatAni-edited scenes are deliberately modified to create additional hazards, which increase testing collision rates. After augmentation and fine-tuning, we observe lower collision rates and more deceleration/stop behavior in these generated hazardous scenes. We use the default DriveLM training setting except for seed 1111, one fine-tuning epoch, learning rate $1 \times 10^{-6}$, and batch size 1. These results should be interpreted as a preliminary evaluation on generated hazardous scenes rather than a general safety guarantee. Additional protocol details are provided in Appendix S7; see the case in Appendix S8 and the supplementary video.

## 5    Conclusion

We propose ChatAni, a language-driven framework for interactive multi-actor animation generation in street scenes. We introduce PedAnimator, a unified control policy for physically plausible pedestrian animation generation across multiple tasks, enabling fine-grained interactions. PedAnimator uses body-masked AMP to improve the plausibility of action generation while employing task masking to implement a unified approach to various control methods. We introduce VehAnimator, a kinematics-based vehicle control policy with a history-aware design to generate kinematically compliant vehicle animation. VehAnimator uses a learning-based approach based on a bicycle model to improve vehicle trajectory tracking and temporal consistency. ChatAni also utilizes multi-LLM-agent role-playing to enable interaction-aware high-level planning under language commands. In the future, we plan to extend ChatAni to more diverse traffic participants, such as cyclists, broader open-vocabulary interaction skills, and more fine-grained contact physics for pedestrian-vehicle collision scenarios.

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

---

**Algorithm 1** Two-Stage Unified Training for PedAnimator

---

**Require:** Human motion dataset, PPO hyperparameters, Pretrained latent decoder.
**Require:** $N_{total}$: Total training steps, $N_{amp}$: Step threshold for AMP activation.
 1: Initialize control policy $\pi_\theta$, value function $V_\phi$, discriminator $D_\psi$.
 2: $global\_step \leftarrow 0$
 3: **while** $global\_step < N_{total}$ **do**
 4:     Sample task $T \in \{$Following, Imitation, Interaction$\}$.
 5:     Initialize state $\mathcal{S}^p$ and apply task mask (activate relevant states, mask others).
 6:     **while** not done and step $<$ max_episode_steps **do**
 7:         Get observation $\mathcal{S}^p = (\mathcal{S}^p_{traj}, \mathcal{S}^p_{mo}, \mathcal{S}^p_{tar}, \mathcal{S}^p_{prop})$.
 8:         Sample latent feature $\mathbf{f_{action}} \sim \pi_\theta(\cdot|\mathcal{S}^p)$ and decode into control signals.
 9:         Step physics simulator, observe next state $\mathcal{S}^p_{next}$ and task reward $R_{task}$.
                                                    ▷ Stage Transition based on fixed training steps
10:         **if** $global\_step \geq N_{amp}$ **then**
11:             Calculate discrimination reward $R_{disc}$ using $D_\psi$ with body mask.
12:             $R^p \leftarrow 0.5R_{task} + 0.5R_{disc}$
13:         **else**
14:             $R^p \leftarrow R_{task}$                                     ▷ Warm-up phase (AMP-free)
15:         **end if**
16:         Store transition in buffer.
17:         $global\_step \leftarrow global\_step + 1$
18:     **end while**
19:     Update $\pi_\theta$ and $V_\phi$ using PPO objectives.
20:     **if** $global\_step \geq N_{amp}$ **then**
21:         Update $D_\psi$ using human reference clips.
22:     **end if**
23: **end while**

---

## S1 Algorithm Pseudocode

To provide a more comprehensive and technical understanding of the proposed ChatAni framework, we present the detailed pseudocode for its core training and inference pipelines. The algorithmic implementations are organized into three primary parts:

- **PedAnimator Training:** Algorithm 1 details the two-stage unified training strategy for the pedestrian controller. It illustrates the transition from the AMP-free warm-up phase to the AMP-enhanced phase based on fixed training steps, along with the application of task masking for unified multi-task learning.

- **VehAnimator Training:** Algorithm 2 outlines the history-aware training procedure for the vehicle controller. It demonstrates the end-to-end TD3 optimization process within a randomized mixed environment, strictly governed by the kinematic bicycle model.

- **System Inference Pipeline:** Algorithm 3 summarizes the holistic multi-LLM-agent role-playing scheme during execution. It covers the Oracle Agent's initialization, the inter-agent communication mechanism (strictly bounded to a maximum of 5 rounds to ensure system efficiency), and the final dispatch to specialized low-level animators.

## S2 Supplementary User Study

This section provides additional quantitative user experiment results to validate the relevant modules using clear numerical data.

---

**Algorithm 2** History-Aware Training for VehAnimator

---

**Require:** TD3 hyperparameters, vehicle parameters $\Theta^v$.
1: Initialize Actor $\mu_\theta$ and Critic networks $Q_{\phi_1}, Q_{\phi_2}$ with target networks.
2: **for** each episode **do**
3:     Initialize randomized mixed environment (varying speeds, random obstacles).
4:     Sample target trajectory and generate initial state $\mathcal{S}^v = (\mathbf{P}^v, \mathbf{V}^v, \Theta^v)$.
5:     **while** not done and step < max_episode_steps **do**
6:         Select action $\mathcal{A}^v_{policy} \leftarrow \mu_\theta(\mathcal{S}^v)$ + exploration noise.
7:         Apply temporal filtering: $\mathcal{A}^v_t = \alpha \mathcal{A}^v_{t-1} + (1 - \alpha)\mathcal{A}^v_{policy}$.
8:         Execute action (delta steering $\Delta\delta$, scalar acceleration $a$) via Bicycle Model.
9:         Apply the bicycle-model transition and observe next state $\mathcal{S}^v_{next}$.
10:         Calculate reward $R^v \leftarrow R^v_{pos} + R^v_{act} + R^v_{obs}$ (if obstacles exist).
11:         Store transition in replay buffer $\mathcal{B}$.
12:         Sample mini-batch from $\mathcal{B}$ and update Critic networks $Q_{\phi_1}, Q_{\phi_2}$.
13:         **if** step mod policy_delay == 0 **then**
14:             Update Actor $\mu_\theta$ using deterministic policy gradient.
15:             Soft update target networks.
16:         **end if**
17:     **end while**
18: **end for**

---

### S2.1 User Study of VehAnimator

Table S1: User preference of VehAnimator and other baselines. Users are asked to choose the most visually plausible motion, and the preference rates are reported.

| Method | Without tracking | PP | Xu's method | Ours |
|---|---|---|---|---|
| Preference | 0 | 0.12 | 0.29 | 0.59 |

We generated 20 trajectory sets, each containing anonymized outputs from the compared methods, and asked 10 participants to select the most visually plausible outcome in each set. This resulted in 200 preference votes. The output order is randomized for each question, and method identities are hidden from participants. The results in Tab. S1 indicate that the majority of participants preferred our method's results. Furthermore, the trajectories generated without tracking were consistently evaluated as less plausible.

### S2.2 User Study of PedAnimator

Table S2: User preference of PedAnimator and other baselines. Users are asked to choose the most visually plausible motion, and the preference rates are reported.

| Methods | PPO | AMP | PULSE | Ours |
|---|---|---|---|---|
| Preference | 0.04 | 0.18 | 0.22 | 0.56 |

We provide comprehensive visual quality comparisons in the supplementary video. Additionally, we conduct a user study with 15 interaction scenarios, asking 10 participants to select the higher-quality motion from anonymized, randomly ordered method outputs. This resulted in 150 preference votes. The results in Tab. S2 indicate that our approach achieves a higher user preference compared to the other methods.

---

**Algorithm 3** Multi-LLM-Agent System Inference Pipeline

---

**Require:** User natural language instruction, Scene map graph $\mathcal{G} = (\mathcal{N}, \mathcal{E})$.

 1: **Oracle Agent:** Parse instruction, output initialization info, interaction groups, and schedules.
 2: Initialize Actor Agents for all traffic participants.
 3: **for** each Actor Agent **do**
 4:     Receive agent type, initial trajectory, and behavioral descriptions.
 5: **end for**
 6: **High-level Planning:**
 7: **for** each schedule step defined by Oracle **do**
 8:     $round \leftarrow 0$
 9:     **while** $round < 5$ **do**                           ▷ Max 5 rounds to prevent infinite loops
10:         **Inter-agent Communication:** Agents exchange target info (e.g., state, intent).
11:         **if** consensus reached on interaction details **then**
12:             **break**
13:         **end if**
14:         $round \leftarrow round + 1$
15:     **end while**
16:     Retrieve map keypoints from $\mathcal{G}$ and derive interaction keypoints.
17:     Generate continuous trajectory via Bézier curve interpolation.
18:     (For Pedestrians) Convert descriptions to upper-body motion via Text2Motion.
19: **end for**
20: **Low-level Execution:**
21: **while** simulation not finished **do**
22:     **for** each participant **do**
23:         **if** type == Pedestrian **then**
24:             Get action via PedAnimator given planned trajectory and target state.
25:         **else if** type == Vehicle **then**
26:             Get action via VehAnimator given planned trajectory and dynamic parameters.
27:         **end if**
28:     **end for**
29:     Update each participant with its corresponding transition: Isaac Gym for pedestrians and the bicycle model for vehicles.
30: **end while**

---

### S2.3 Preliminary User Study of the System

Given the lack of directly comparable prior work, we conducted a preliminary user study to evaluate our approach before the larger full-system evaluation in Tab. 2. We generated 20 distinct final results alongside their corresponding commands, and asked 10 participants to assess whether the animations were visually plausible and aligned with the input requirements. This resulted in 200 binary judgments. The sample order is randomized for each participant. The results indicate that in 69% of the judgments, participants agreed the commands were matched and the resulting animations were visually plausible.

## S3 Ethics, Reproducibility Statement, and Usage of LLMs

All related human participants involved in the user study provided consent, and there are no additional ethical concerns.

**Reproducibility and artifact availability.** To support reproducibility, the supplementary materials include the training code and configuration files for the main experiments, selected evaluation code, the high-level planning design and implementation, prompt templates, model settings, pseudocode, and detailed evaluation protocols. The full codebase is being curated and organized for public release. Due to the size of the generated data and trained checkpoints, these artifacts are not bundled with the current submission.

We plan to release the curated full codebase, generated data, and trained checkpoints through a public repository, subject to storage constraints and third-party dataset licenses. Accordingly, we avoid claiming full reproducibility in the current submission and instead specify the artifacts currently provided and the remaining release plan.

**Usage of LLMs.** This work uses GPT-4 Achiam et al. (2023) as a core high-level planning component in the proposed multi-LLM-agent framework. During inference, GPT-4 is prompted as Oracle and Actor agents to parse user instructions, assign participant roles, infer interaction groups and behaviors, and generate JSON-formatted schedules and keypoint instructions. These outputs are then converted by deterministic tools into structured plans that are executed by PedAnimator and VehAnimator. GPT-4 is not used to train the low-level PedAnimator or VehAnimator policies, and it is not used to evaluate generated results. Separately, LLM-based writing tools may have been used for minor language polishing, with all technical content, experiments, and claims checked by the authors.

## S4 Details of PedAnimator

### S4.1 PedAnimator State Details

The components of the humanoid's proprioception $S_{prop}^p$ are as follows: joint positions $\mathbf{j} \in \mathbb{R}^{24 \times 3}$, rotations $\mathbf{q} \in \mathbb{R}^{24 \times 6}$, linear velocities $\mathbf{v} \in \mathbb{R}^{24 \times 3}$, and angular velocities $\omega \in \mathbb{R}^{24 \times 3}$ Wang et al. (2024). These components are normalized with respect to the agent's heading and root position in our simulator. The rotation $\mathbf{q}$ is represented using a 6-degree-of-freedom rotation representation. $S_{prop}^p$, together with the trajectory state $S_{traj}^p$, the motion state $S_{mo}^p$ to be mimicked, and the target state $S_{tar}^p$ for interaction, forms the complete observation. During task masking, task-irrelevant states are set to zero according to the active task, and specific joints in the motion state can also be masked when necessary. For example, in our experiments, only upper-body-related motion states are retained for the corresponding control condition.

### S4.2 Task-related Reward and Training Details

**Reward designs.** For trajectory following tasks Rempe et al. (2023), the reward is defined as $R_{\text{trajectory}} = e^{-2\|\hat{p}_t - p_t\|}$, where $\hat{p}_t$ is the target position to be followed at time $t$, and $p_t$ is the current character position. For motion imitation tasks Luo et al. (2023), $R_{\text{imitation}} = e^{-100\|\hat{j}_{pos}^t - j_{pos}^t\| \odot m^t} + e^{-10\|\hat{j}_{rot}^t - j_{rot}^t\| \odot m^t} + e^{-0.1\|\hat{j}_{vel}^t - j_{vel}^t\| \odot m^t} + e^{-0.1\|\hat{j}_{\omega}^t - j_{\omega}^t\| \odot m^t}$, where ˆ indicates the motion states to be imitated, and $m^t$ is the joint mask used for imitation. In our experiments, this mask selects upper-body joints for the corresponding control condition. Different interaction tasks require distinct reward designs Peng et al. (2022); here, we present three sample tasks. For pushing, $R_{\text{pushing}} = 1 - u^{up} \cdot u^{tar}$, where $u^{up}$ is the global up vector and $u^{tar}$ is the target's up vector. For patting, $R_{\text{patting}} = e^{-\|p^{rh} - c\|}$, where $p^{rh}$ is the right-hand position and $c$ is the target contact position. For walking with an arm around another character's shoulder, $R_{\text{walking\_shoulder}} = e^{-2\|\hat{p}_t - p_t\|} + e^{-\|p^{rh} - c\|}$, which combines trajectory following with contact-position tracking. The final reward is calculated as $R = 0.5 \cdot R_{\text{disc}} + 0.5 \cdot R_{\text{task}}$.

**Interaction process.** The specific execution process for the three interaction tasks can be understood as follows: i) pushing, where the goal is to push the interaction object over, ii) patting, where the task is to gently tap a specific part of the interaction object (e.g., the shoulder) without knocking it over, and iii) walking with arm around another's shoulder, where the agent walks along a specific path while keeping the arm in contact with a specific location on the interaction object (e.g., the shoulder). During the training of the interaction tasks, the interaction object is replaced with a box in the physical environment to facilitate more stable training. However, during testing, the interaction object is replaced with another character in the simulation environment, and physical collisions and interactions are present, leading to the final output in the test phase.

**Failure recovery and vehicle-pedestrian contact.** To improve the recovery behavior of PedAnimator, specifically to help it handle external disturbances caused by physical collisions, we incorporate recovery during its training Luo et al. (2023). In this approach, the human pose is initialized in a collapsed or otherwise unstable standing state. Training is then conducted from this initial state, allowing the policy to

learn how to recover from failure. This helps the policy respond to physical collisions and interactions in the evaluated scenarios. Without this, the policy might fail to continue the action after even minor disturbances.

Vehicle-pedestrian contact scenarios are approximated in Isaac Gym with a vehicle-like rigid box that interacts with the pedestrian body. The resulting pedestrian motion is read out from the simulated humanoid state. The vehicle motion in these scenarios is specified by the high-level planning process, including the velocity changes associated with the contact event.

### S4.3 Network Architecture

For the PPO, AMP, and PULSE interaction-control baselines retrained in our shared environment, we use an MLP policy with hidden layers of 2048 and 1024 units. The final output is directed to either the latent space or the PD control signal, depending on whether hierarchical control is employed. The remaining network components, such as the discriminator, value network, control frequency (30Hz), and hyperparameters used for training, are consistent with those adopted in Pacer+ Wang et al. (2024). All training and testing are conducted on an NVIDIA 4090. The entire training process requires approximately 20 hours to fully converge.

### S4.4 Evaluation Details

**Following and imitation.** For the following and imitation tasks, we adopt the same computation methods as those used in Pacer+ Wang et al. (2024). The calculations of FID and diversity are performed using the same manual feature extraction approach as in Pacer+, with 1000 segments selected from the AMASS dataset for FID computation. For the low-speed l-FID and l-diversity, we also follow Pacer+ by testing on instances where the speed is less than 1 m/s.

**Interaction tasks.** For the three interaction-related tasks: i) pushing, the success criterion is whether the object is pushed over within the specified timestep (with a tilt along the z-axis greater than 30°), and no part of the body other than the hands is in contact with the target; ii) patting, the success criterion is whether, within the specified timestep, the right hand is within 5 cm of the target's specific location and remains there for at least 50 timesteps, with no other part of the body in contact with the target except the hands; iii) walking with arm around another's shoulder, the success criterion is that the maximum deviation from the reference trajectory is no greater than 10 cm, and the right hand is within 5 cm of the target's specific location for at least 150 timesteps. In the interaction ablation study, we recruit 15 participants to evaluate 30 dynamic sequence comparisons, resulting in 450 preference votes. The compared outputs are anonymized and randomly ordered, and participants select the one they consider to have higher quality.

### S4.5 Rendering Details

We utilize the rendering pipeline from ChatSim Wei et al. (2024), employing Blender's Cycles as the rendering engine. Background rendering and HDRI lighting from ChatSim are applied to the scene. The human model is based on SMPL Loper et al. (2023), with the corresponding mesh initialized, and rendered using skin and clothing textures provided in the Bedlam dataset Black et al. (2023).

### S4.6 Supplementary Experiments

We provide more comprehensive visualization results to compare different aspects of PedAnimator's characteristics. All comparisons, along with animation results, can also be observed in the supplementary video material.

**Ablation study** As shown in Table S3, we conduct ablation studies on interaction tasks to verify the effectiveness of task embedding (TE), hierarchical control (HC), and body-masked AMP (M. AMP). The results indicate that, without task embedding, PedAnimator struggles to accurately identify the required interaction task, leading to difficulty in task completion. When body-masked AMP is removed, similar to AMP, the discrimination reward lacks a task completion reference. In this situation, the results can be

Imitation and following comparison

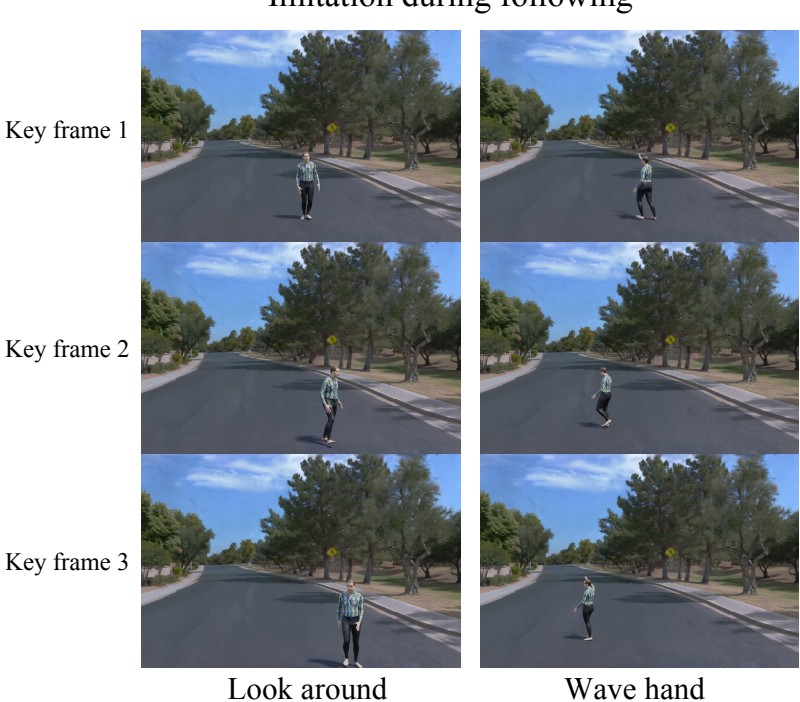

Figure S1: Comparison of imitation and following.

Imitation during following

Figure S2: Imitation during following.

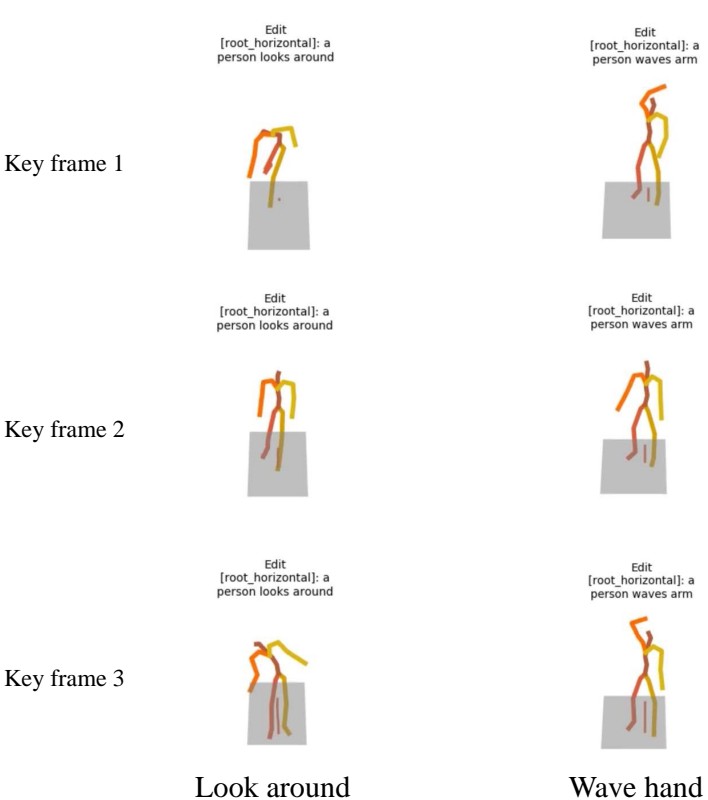

Figure S3: Failure of PriorMDM Shafir et al. (2023) with action specification during following.

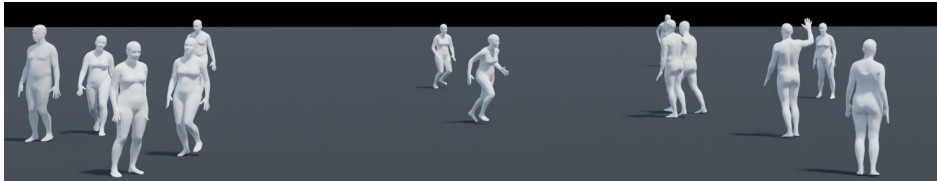

Figure S4: Animations generated by PedAnimator with MoCap, T2M, and interaction target control. PedAnimator generates diverse and visually plausible animations under various control signals.

| TE | HC | M. AMP | Success rate | User preference |
|----|----|--------|--------------|-----------------|
| × | ✓ | ✓ | 0.403 | 0.120 |
| ✓ | ✓ | × | 0.186 | 0.069 |
| ✓ | × | ✓ | 0.948 | 0.344 |
| ✓ | ✓ | ✓ | **0.977** | **0.467** |

Table S3: Ablation study of PedAnimator. Each design contributes to the final results, and the complete PedAnimator obtains the strongest results among these variants.

merely close to the reference data clip without the ability to successfully execute the intended task. Without hierarchical control, the absence of action priors slows the training process and reduces the success rates.

**Imitation and following comparison.** As shown in Fig. S1, we compare the visual effects of Pacer+ during the imitation and following processes. In this example, hierarchical control helps PedAnimator produce smoother transitions between different tasks than the AMP-prior baseline.

**Results of imitation during following.** As shown in Fig. S2, we also provide the results of PedAnimator performing both imitation and following simultaneously. In the case of following a specific trajectory, the upper body performs actions such as looking around and waving, showing PedAnimator's capability to handle both imitation and following tasks concurrently.

**Interaction task comparison.** As shown in Fig. S6, Fig. S7, and Fig. S8, we compare the performance of AMP, PPO, PULSE, and our method across three interaction tasks. In our visual comparisons, PPO and PULSE are able to complete the tasks under the given conditions but often produce less natural movements. AMP, on the other hand, can only approximate the reference in the discriminator (walking or running), and does not complete the interaction tasks in our tests. In contrast, PedAnimator completes the tasks while generating more natural and human-like results.

**Multiple pedestrians visualization.** Fig. S4 shows PedAnimator's generation capabilities under various control modalities, including MoCap, Text2Motion-driven upper-body control, trajectory control, and interactive behavior control. The system synthesizes diverse and visually plausible animations through planned control schemes.

### S4.7 Further Discussion of Kinematics Methods

For kinematics-based methods, some approaches can achieve following and motion specification, but they are not designed to handle interaction-related tasks. Additionally, for following and motion specification tasks, these methods often suffer from overfitting to the dataset, leading to suboptimal performance. As shown in S3, the results include noticeable sliding steps and unnatural movements.

## S5 Details of VehAnimator

### S5.1 Network Architecture, Bicycle-model Parameters, and Training Details

All networks in VehAnimator are implemented as MLPs. The policy network consists of layers with dimensions 256, 256, 128, 128, 64, and 64, while the value network has layers with dimensions 1024, 512, 256, and

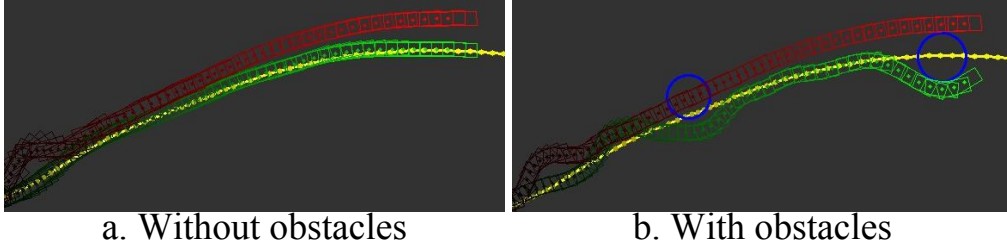

| a. Without obstacles | b. With obstacles |

Figure S5: Comparison of vehicle animation generation. Green boxes are our results and red boxes are Xu et al.'s Xu & Yu (2023). Yellow lines are the planned trajectory and blue circles are obstacles. VehAnimator shows lower tracking error and obstacle avoidance in this example.

| Filt. | Action spa. | His. state | Position error | Velocity error |
|:-----:|:-----------:|:----------:|:--------------:|:--------------:|
| ✗ | ✗ | ✗ | 0.138 | 0.114 |
| ✓ | ✗ | ✗ | 0.132 | 0.111 |
| ✓ | ✓ | ✗ | 0.115 | 0.092 |
| ✓ | ✓ | ✓ | **0.106** | **0.088** |

Table S4: VehAnimator ablation study. Each component contributes to lower errors in the final result.

128. During training, the parameters of the bicycle model (L, W, $l_f$, $l_r$) are set to two configurations: (2.7, 1.8, 0.9, 0.9) and (6.1, 2.5, 2.3, 2.0), mixed to accommodate vehicles of varying sizes. These parameters can be adjusted as needed based on specific requirements, with the two configurations provided here serving as examples. The training environment for VehAnimator was implemented through our custom codebase, with the full training process requiring approximately 10 hours on a single NVIDIA RTX 4090 GPU.

### S5.2 Obstacle state

The states of obstacles are composed of their orientation and distance relative to the vehicle's own coordinate system (considering the radius of the obstacles). The state vector is initialized with a maximum number of observable obstacles. The vector is then populated with the specific identifiers of the actual obstacles, and any remaining entries are masked. Obstacles that are too far away are directly excluded, meaning their states are not considered, and they do not contribute to the reward calculation. The distance threshold for exclusion is set to 10 in the experiment.

### S5.3 Supplementary Experiments

**Ablation study.** We also conduct an ablation study in Table S4, examining the effects of action filtering (Filt.), the composition of the action space (Action spa.), and the inclusion of historical states (His. state). The ablation results validate the effectiveness of each design.

**Visualization.** Furthermore, visual experiments in Fig. S5 illustrate that our method achieves lower tracking error and obstacle avoidance in scenarios with obstacles.

**Sensitivity to trajectory noise.** We also provide the results using LQR Li & Todorov (2004) in Table S5. LQR can generate vehicle animations from planned trajectories to some extent. However, because the planned trajectory lacks relevant constraints, it may contain unreasonable turns or abrupt changes, causing LQR to often produce less stable results. Furthermore, we evaluate the sensitivity of vehicle animation generation by adding noise with a mean of 0 and variance of $\sigma$ to the planned trajectory. As shown in Table S6, LQR is highly sensitive to noise and often produces larger errors under its influence, while the other methods are relatively less affected.

**Visualization for effectiveness.** In the supplementary video, we provide results comparing the planned trajectories without using VehAnimator and with VehAnimator. In these examples, without the involvement of VehAnimator, the animation generated by simply calculating heading between consecutive frames of the directly planned trajectory is less natural, exhibiting noticeable tail swings and abrupt changes. In

| Methods/Speed | 0 | 5 | 10 | 20 |
|---|---|---|---|---|
| LQR Li & Todorov (2004) | 0.074/0.058 | 0.086/0.070 | 0.092/0.079 | 0.125/0.108 |

Table S5: Position/velocity error of LQR.

| $\sigma$ | PP Craig Coulter (1992) | Xu et al. Xu & Yu (2023) | LQR Li & Todorov (2004) | ours |
|---|---|---|---|---|
| 0.00 | 0.162/0.142 | 0.095/0.077 | 0.092/0.079 | **0.077/0.054** |
| 0.01 | 0.168/0.147 | 0.098/0.080 | 0.322/0.289 | **0.079/0.058** |
| 0.03 | 0.169/0.151 | 0.098/0.079 | 0.568/0.479 | **0.082/0.060** |

Table S6: Vehicle animation generation under Gaussian noise. $\sigma$ indicates the variance of noise.

contrast, the results using VehAnimator are more natural. This demonstrates the utility of VehAnimator, as trajectories that do not satisfy the bicycle-model transition can produce visually implausible vehicle motions.

## S6 Details of High-level Planning

### S6.1 LLM-agent details

We provide the relevant sample prompts for the LLM agent in S10 and S11. The agents produce JSON-formatted structured outputs, with selected text fields containing natural-language behavior descriptions. Corresponding deterministic follow-up functions convert these outputs into the required data structures.

Similar to the rendering process, we also use the Waymo Open Dataset Sun et al. (2020) as the planning dataset in all experiments, and the final results are presented based on this dataset.

### S6.2 Different LLMs

We further validated the impact of different LLMs on the results in S7. We conducted experiments using GPT-3.5 Ouyang et al. (2022) and Llama-3 Dubey et al. (2024) 70B (smaller models struggle to accurately execute the instructions). All other experimental settings remained consistent with those in the main text. It is evident that while other LLMs can handle the task to some extent, GPT-4 Achiam et al. (2023) demonstrates the most accurate understanding and decomposition of the instructions.

### S6.3 Collision handling

In the high-level planning process, we also designed a collision handling mechanism to avoid unintended collisions. Specifically, during the trajectory generation, collision detection is performed, and when a collision is detected, a velocity adjustment function is applied to modify the speed of one of the agents. This velocity adjustment function uses a nonlinear mapping to combine the original planned result with an interpolated trajectory, leading to a planning result with different speeds. We evaluated the probability of collisions across 50 generated samples, which is calculated by dividing the number of vehicles that experienced a collision by the total number of vehicles. As shown in S8, the collision rate for all methods remains low, and ChatAni also achieves a low collision rate while incorporating collision handling.

### S6.4 Evaluation for control accuracy

To evaluate command compliance, we introduce three quantitative metrics: sequence accuracy (order), positioning error (position), and velocity error (speed). Table S9 summarizes the performance of the evaluated methods on a test set of 100 commands. The results indicate that LCTGen shows higher error margins in position and speed, along with lower sequence accuracy, which aligns with our previous findings. ChatSim provides improved control accuracy. In comparison, our method achieves the highest sequence accuracy (0.93) and the lowest errors in both positioning (0.28) and velocity (0.25) among the evaluated methods.

## Pushing

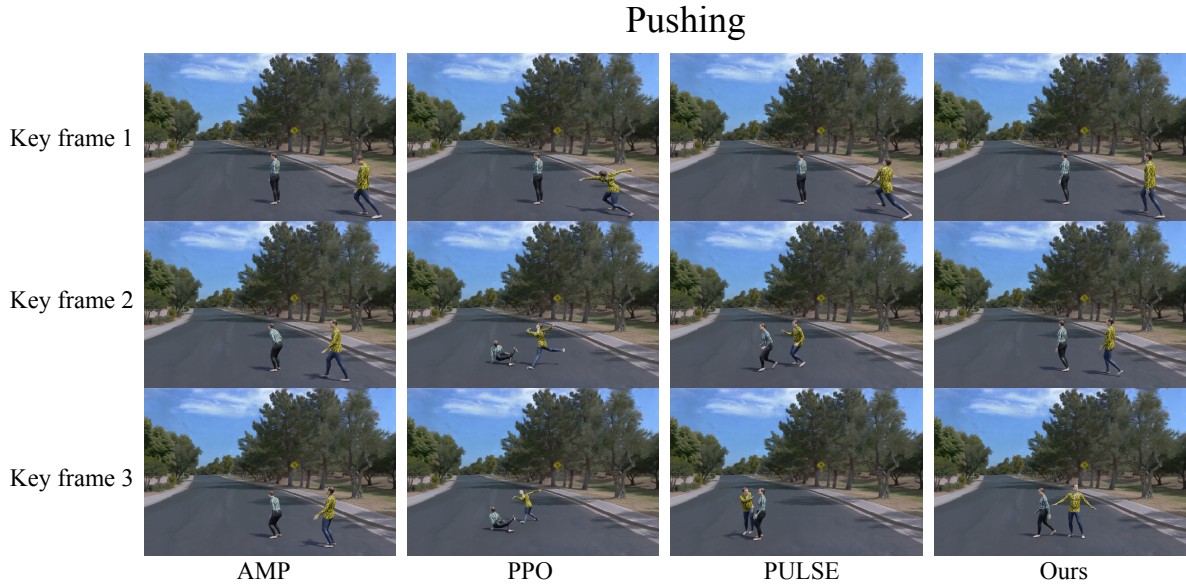

Figure S6: Comparison of pushing.

## Patting

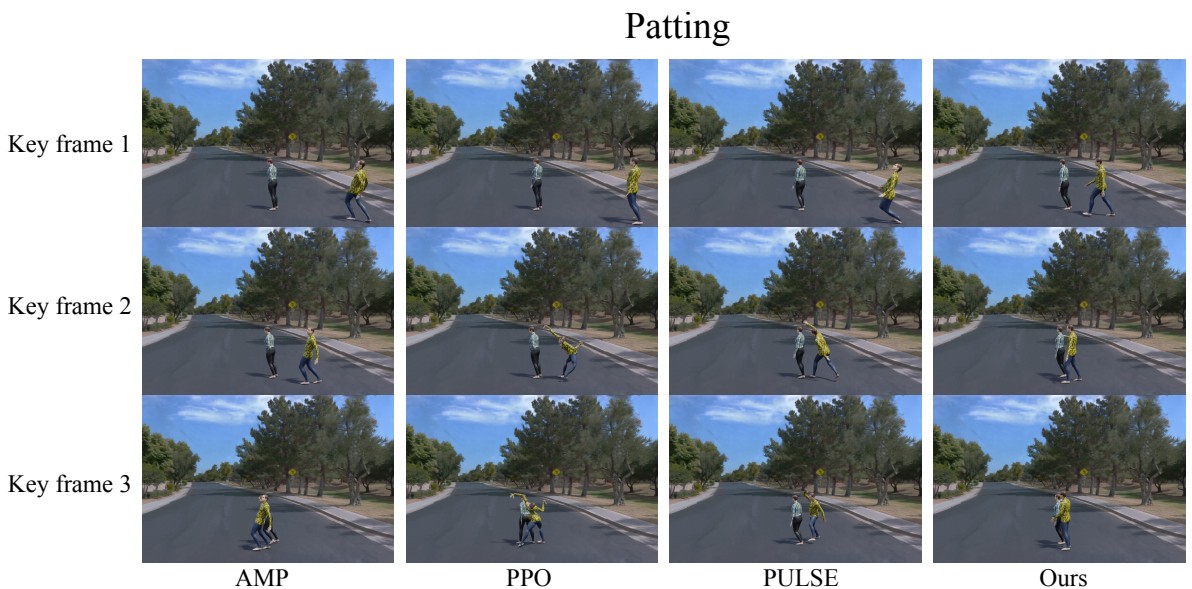

Figure S7: Comparison of patting.

| Methods | Language command category | | | Within road |
|---|---|---|---|---|
| | single | interaction | compound | |
| Ours-Llama3 | 0.885 | 0.742 | 0.812 | 0.920 |
| Ours-GPT3.5 | 0.854 | 0.738 | 0.834 | 0.915 |
| Ours-GPT4 | **0.952** | **0.883** | **0.896** | **0.935** |

Table S7: High-level planning evaluation for different LLMs.

| | ChatSim Wei et al. (2024) | LCTGen Tan et al. (2023) | Ours |
|---|---|---|---|
| Collision rate | 0.149 | 0.092 | 0.067 |

Table S8: Collision rate of high-level planning

## S7 Downstream Evaluation Protocol

We provide additional protocol details for the downstream evaluations in Tab. S10. These experiments use a fixed seed of 1111 and are intended to evaluate the potential utility of generated data under controlled settings. For prediction experiments, synthetic samples are generated only from the training split, and held-out evaluation splits are not used during augmentation. For the DriveLM evaluation, the 30 evaluation scenes are excluded from the 3000-frame fine-tuning set, and we check that there is no scene/frame overlap.

For the DriveLM evaluation, a collision is counted when the ego vehicle collides with any generated or existing pedestrian or vehicle. The reported collision rates therefore measure behavior on this generated hazardous-scene benchmark and should not be interpreted as a general guarantee of autonomous-driving safety. Across 300 trials per condition, the aggregate collision rates and Wilson 95% confidence intervals are 8.6% [5.9, 12.3] for the original scenes, 75.3% [70.1, 79.8] for the ChatAni-edited scenes, and 17.3% [13.4, 22.0] after augmented fine-tuning.

## S8 Corner case for VLM

As shown in Fig. S9, we demonstrate a corner case for Visual-Language Model (VLM)-related experiments, with the command: "Add a stationary car and have a pedestrian walk out from behind it." In this scenario, the ego vehicle initially fails to detect the pedestrian while moving forward. However, as it approaches the stationary car, it comes dangerously close to the pedestrian, leading to a collision–a visibility-induced corner case. Without fine-tuning, collisions occur frequently due to the system's inability to anticipate the pedestrian's presence. After fine-tuning in our generated hazardous-scene setting, the model more often outputs deceleration when large blind zones are detected in the field of view, reducing collision probability in this evaluated scenario. The related video can be found in the supplementary video.

## S9 User study details

We conducted user studies via questionnaire format, distributing surveys containing samples from different experimental sources for participants to select the option that best meets requirements or visual preferences.

Table S9: Evaluation of command compliance across different methods.

| Method | LCTGen | ChatSim | Ours |
|---|---|---|---|
| Order (↑) | 0.18 | 0.87 | **0.93** |
| Position (↓) | 2.83 | 0.64 | **0.28** |
| Speed (↓) | 1.98 | 0.46 | **0.25** |

## Walking with arm around another's shoulder

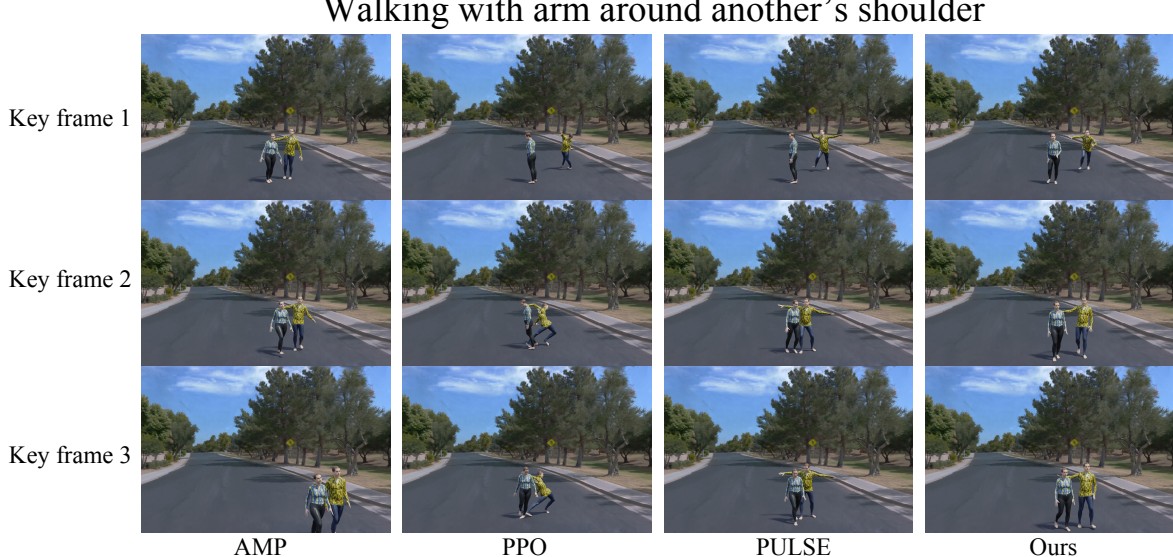

Key frame 1

Key frame 2

Key frame 3

AMP      PPO      PULSE      Ours

Figure S8: Comparison of walking with arm around another one's shoulder.

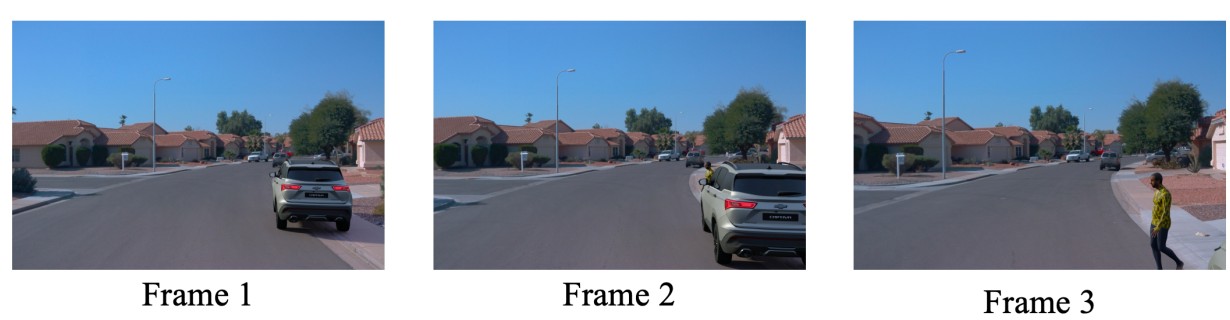

Frame 1      Frame 2      Frame 3

Figure S9: Corner case sample.

Throughout the paper, *participants* denotes unique human evaluators within a study, while *judgments* or *votes* denotes the total number of collected responses. For preference studies, method identities are hidden, outputs are shown with anonymized labels, and the display order is randomized for each question. Preference rates are calculated as the number of valid votes for each method divided by the total number of valid votes. For binary studies, participants provide yes/no judgments for each prompt-result pair, and we report the percentage of positive judgments. For command-following studies, the corresponding prompt is visible to participants. Specific implementation details are summarized in Tab. S11, with an interface example provided in Figure S12.

### S9.1 Statistical Analysis

For the main binary and preference metrics reported in the main paper, we provide Wilson 95% confidence intervals. In the high-level planning evaluation in Tab. 6, each table entry aggregates $5 \times 26 \times 100$ responses, and the main paper reports the corresponding Wilson confidence-interval half-widths. In the end-to-end full-system evaluation in Tab. 2, each category-level metric is computed from 100 judgments, and each overall metric is computed from 600 judgments for that criterion. Across the five criteria, the evaluation contains 3,000 binary judgments. Each PedAnimator interaction success rate in Tab. 4 is computed from 1,000 evaluation episodes and reported with a Wilson 95% confidence interval. The DriveLM evaluation contains

**Oracle agent prompt.**

I have a requirement for analyzing a scenario. I will provide you with a requirement, and I need your help to break it down into three pieces of information: (1) identify all the agents, (2) initialize each agent's state, (3) formulate a text for each agent. You should provide the information in JSON format. Note that your output should only include the JSON format of the information, not the analysis process.

(1) Identify all the agents: This means you need to extract all the agents from the scenario based on the input text. The text may be quite lengthy, so you can locate the agents by identifying the nouns, which may assist you in this task. The agents must be the objects involved in autonomous driving. The format should be as follows: agent_list = {'0': 'agent_name_0', '1': 'agent_name_1', ...}. The keys (e.g., '0', '1', etc.) represent unique agent IDs, which you should assign starting from '0'. The agents should only include vehicles like cars, pedestrians, trucks, and buses, and should not include static objects like trees or buildings. Ensure that each agent is given a distinct name. For example, in the sentence 'car a wants to overtake car b', the nouns are 'car a' and 'car b'. Both are objects in autonomous driving scenarios. Therefore, the agent_list should be formatted as follows: agent_list = {'0': 'car_a', '1': 'car_b'}. Pedestrians can be represented with identifiers like 'ped_0' and may share numbering with entities such as 'car_0'.

(2) Initialize each agent's state. In this task, you need to determine four aspects for each agent: 1. Agent type, 2. Movement, 3. Speed. You can use the results from task (1) to complete this task. Provide the initial state of each agent in a list, formatted in JSON. Most time the speed is bigger than 0. For example, the initial states should be defined as follows: init-states = ['agent_id': '0', 'agent_type': 'car', 'movement': 'overtaking', 'speed': 60, 'agent_id': '1', 'agent_type': 'car', 'movement': 'straight', 'speed': 30]. If one car intends to overtake another, it should ideally be at least twice as fast as the car it is overtaking. The initial state must include the agent type, movement,and speed.

Agent type should in [pedestrian,vehicle], action should in ['static', 'straight','pull over', 'turn over','overtake','turn left','turn right','straight left','straight right'] if it's a vehicle, and in ['static','crossing','straight'] if it is a pedestrian.

(3) Formulate a text for each agent. As an omniscient observer, you should instruct each agent on their actions through a text. The text should contain two pieces of information: 1. The agent type, 2. The agent's intention. You need to provide this in a JSON format. guide_texts = '0': 'text1', '1': 'text2', '2': 'text3', ... where the key is the agent's ID and the value is the text. The agent's name should be consistent with those in the agent_list.

Your answer should be in a JSON format,and must include the three information:agent_list,init-state,guide_texts.And init-state must include the agent type,movement and speed.

Figure S10: Oracle agent prompt.

**Actor agent prompt.**

Now,you are an agent in the autonomous driving scenario.I will give you a text,and agent_list, describing who you are and what you need to do. Note that your output should not include your analysis process, only the JSON format of the information you provide.

I need you to analyze the text and give me the four information of the ego agent to describe what type of lane the agent should be in: (1)depend (2) speed change (3) keypoints list (4) behavior. Note that you only need to give the information of the ego agent,not the other agents.

(1) depend. You need to determine the depend of the agent according to the agent's intention. And give it like [depend_agent_id,depend_type].You can get the depend_agent_id from the agent_list. Depend_type should in ['end','start','trajectory','None'].'end' represents the ego agent's end point is the depend agent's end point, 'start' represents the ego agent's start point is the depend agent's start point, 'trajectory' represents the ego agent's whole trajectory is the depend agent's start point. If it has no depend, you should give it [-1,'None'].

(2) speed change. You need to determine the speed change of the agent according to the agent's intention. If you want to speed up, the speed change should be 1. If you want to slow down, the speed change should be -1. If you want to keep the speed, the speed change should be 0. For example, If there is someone nearby, you should slow down.

(3) keypoints list. You need to determine the number of keypoints required for your current behavior, the confirmation method for each keypoint, and their respective parameters, then return a list containing this information. Each keypoint can be confirmed in one of three ways: (1) Map-based (type '0'), with parameters: lane position ('left', 'right', or 'front'), lane type ('centerline' or 'boundary'), and driving direction ('turn left', 'turn right', or 'straight'). (2) Lane-relationship-based (type '1'), with the parameter being the relationship to the previous keypoint ('opposite direction adjacent', 'same direction adjacent', 'adjacent straight', 'adjacent left turn', 'adjacent right turn', 'different type adjacent', or 'opposite boundary'). (3) Agent-based (type '2'), with parameters specifying required information type ('point' or 'trajectory'). The result should be a sequential list of dictionaries where the key is the type ('0', '1', or '2') and the value is the corresponding parameters. For example, a left-turning car might return '['0': 'position': 'front', 'lane_type': 'centerline', 'direction': 'turn left', '1': 'relationship': 'adjacent straight']', while a car in the left lane picking up 'ped_a' might return '['0': 'position': 'left', 'lane_type': 'centerline', 'direction': 'straight', '2': 'info_type': 'point']'. Ensure compliance with parameters, common sense, and traffic regulations, with the first keypoint typically using type '0'.

(4) behavior. You need to determine your behavior. If you are a vehicle, the behavior is "None." If you are a pedestrian, the behavior corresponds to the description provided. There are two scenarios: if your behavior is a specific action such as calling or waving, simply return the text of that action; if the behavior involves interactive actions such as pushing, patting, or walking with an arm around another person, you must return the exact predefined descriptions for these three types of interactions without modification. For example, if you are calling, your behavior is "calling phone."; if you push ped_1, your behavior is "pushing."

Your answer should be in a JSON format, and must include depend, speed change, keypoints list and behavior.

Figure S11: Actor agent prompt.

Table S10: Downstream evaluation protocol. We use matched augmentation budgets and keep the downstream model configuration fixed within each task.

| Task | Seed | Split/evaluation | Augmentation/fine-tuning data | Leakage and comparison check |
|---|---|---|---|---|
| Traffic prediction with MTR | 1111 | 243k scenes from the official Waymo training split; evaluation on the official validation split | 20% additional training data (48k scenes) from LCTGen or ChatAni | Synthetic data generated only from training-split scenes; same MTR configuration and augmentation budget |
| Human motion prediction with HumanMAC | 1111 | Human3.6M split and preprocessing follow HumanMAC | 20% additional training data (2k samples) from Omnicontrol or PedAnimator | Synthetic data generated only from the training split; same HumanMAC configuration and augmentation budget |
| DriveLM hazardous-scene evaluation | 1111 | 30 Waymo training-split scenes reserved for downstream evaluation; 10 trials per scene | 3000 ChatAni-augmented hazardous frames; one epoch, learning rate $1 \times 10^{-4}$, batch size 1 | Evaluation scenes are excluded from fine-tuning; no scene/frame overlap; compare original, ChatAni-edited, and ChatAni-edited after fine-tuning |

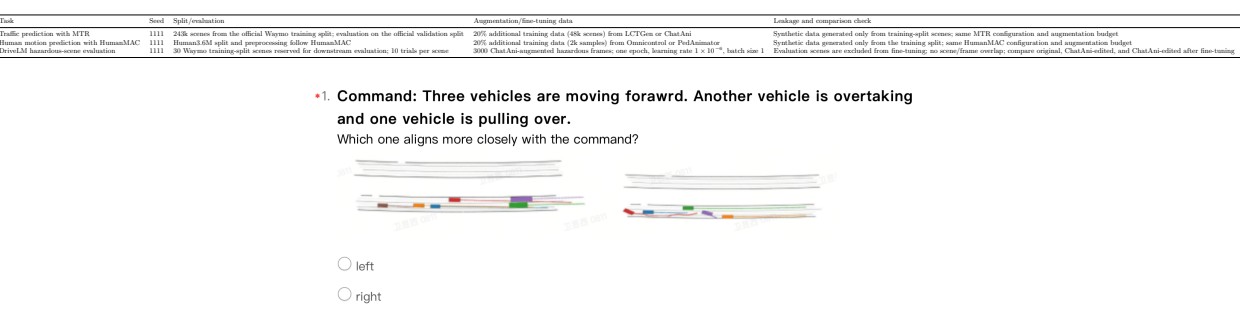

Figure S12: A sample interface of user study.

300 trials per condition and reports aggregate Wilson 95% confidence intervals. For PedAnimator and VehAnimator continuous metrics, we report mean ± standard deviation over three independent runs. When raw paired samples are unavailable, we report these uncertainty estimates rather than adding unsupported paired significance tests from aggregate values.

## S9.2 End-to-end Full-system Evaluation Protocol

For the full-system evaluation in Tab. 2, we generate 120 complete ChatAni results across six categories, with 20 generated scenes for each category. The categories are: (i) single-agent prompts, which specify the behavior of one participant; (ii) vehicle-vehicle prompts, which include interactions such as overtaking, lane switching, and speed adaptation; (iii) pedestrian-vehicle prompts, which include crossing, hailing, collision avoidance, or collision-related events; (iv) pedestrian-pedestrian prompts, which include pushing, chasing, patting, or walking together; (v) compound prompts, which combine multiple participants and interaction requirements; and (vi) crowd/abstract prompts, which describe crowded or semantically abstract traffic situations.

We recruit 20 participants. Each participant evaluates 30 randomly assigned prompt-result pairs, balanced across categories with five samples from each category. Therefore, each generated scene receives five evaluations from different participants, producing 600 prompt-result evaluations in total. Because participants answer five yes/no questions for each pair, the study contains 3,000 binary judgments across the five criteria:

- **Actor coverage:** whether the generated scene contains the participants required by the prompt.

- **Command matching:** whether the generated actors follow the main command requirements.

- **Interaction success:** whether the specified interaction occurs in the generated scene.

- **Map validity:** whether the generated actors remain in semantically valid regions, such as vehicles on drivable areas and pedestrians in plausible walking regions.

- **Scene success:** whether the whole generated scene satisfies the prompt without obvious scene-level failures.

The main paper reports the percentage of positive ratings for each criterion and category.

## S10 Failure Cases and Analysis

### S10.1 High-level Vehicle-planning Failures

Fig. S13 shows two representative failures of high-level vehicle planning. In these bird's-eye-view visualizations, opacity changes from light to dark as time progresses, and the blue vehicle is the target actor assigned

Table S11: Summary of user studies. We distinguish unique participants from total evaluations, judgments, and votes.

| Study | Participants | Items/participant | Total | Protocol |
|---|---|---|---|---|
| PedAnimator following/imitation | 100 | 33 | 3300 | Preference among anonymized, randomized method outputs |
| PedAnimator interaction | 10 | 15 | 150 | Preference among anonymized, randomized method outputs |
| PedAnimator ablation | 15 | 30 | 450 | Preference among anonymized, randomized variant outputs |
| VehAnimator | 10 | 20 | 200 | Preference among anonymized, randomized method outputs |
| High-level planning | 100 | 130 samples per category | 13000 responses per table cell | All participants evaluate every sample; command/road-validity judgments and preference questions |
| Preliminary system study | 10 | 20 | 200 | Binary judgments on prompt-result alignment and visual plausibility |
| End-to-end full-system evaluation | 20 | 30 | 600 evaluations/3000 binary judgments | Five binary criteria per prompt-result evaluation |

the specified behavior. In the first example, the command is "Several vehicles are moving forward; one vehicle accelerates and changes to the left lane to overtake." However, other vehicles obstruct the requested left-lane maneuver. The current coordination mechanism prioritizes its conservative collision-avoidance constraints and instead plans a right-lane change for the target vehicle, thereby completing an overtake but violating the requested direction. In the second example, the command is "Several vehicles are driving; one vehicle pulls over." Surrounding vehicles again obstruct the requested maneuver, and the target vehicle only decelerates rather than completing the pull-over. These examples show that, even when the requested behavior is identified, the current planner may fail to reconcile exact command following with collision avoidance and multi-vehicle coordination when the requested path conflicts with the surrounding traffic configuration.

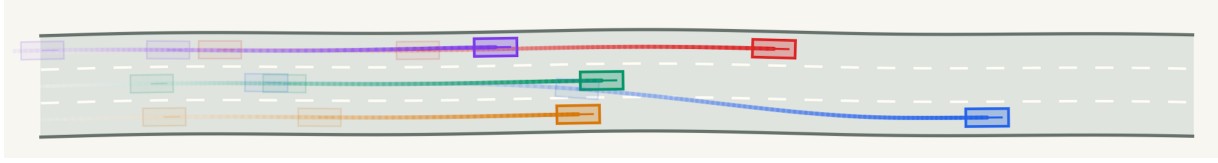

(a) The target vehicle overtakes on the right instead of changing to the requested left lane.

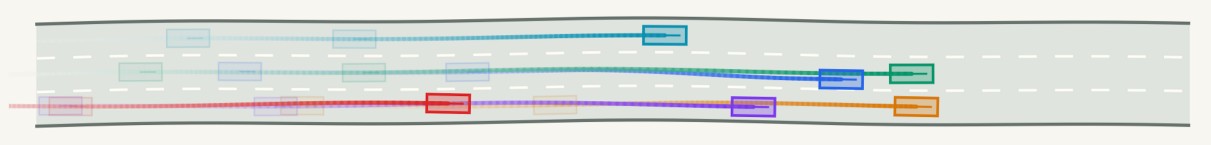

(b) The target vehicle decelerates but does not complete the requested pull-over maneuver.

Figure S13: Representative high-level vehicle-planning failures. Opacity changes from light (earlier states) to dark (later states), and blue denotes the target vehicle. In both cases, surrounding traffic blocks the requested maneuver and the resulting plan does not fully follow the command.

## S10.2  PedAnimator Motion-execution Failures

Fig. S14 shows two component-level PedAnimator failures rather than end-to-end ChatAni failures. The instructions in these examples are given specifically to the PedAnimator motion-generation pipeline and do not pass through the Multi-LLM-Agent planner. MoMask Guo et al. (2024) first converts the instructions into reference motions, from which the upper-body motion conditions are provided to PedAnimator for physically constrained execution. The two instructions are "A pedestrian walks and then squats down" and "A person performs a standing forward jump and then walks." Both motions involve substantial and rapid changes in the character's center of mass. When accurate execution of such a reference motion conflicts with maintaining physical balance, the current PedAnimator tends to prioritize balance. Consequently, the intended actions are not executed accurately and the generated sequences contain abrupt transitions and visually unnatural poses. These cases motivate a controller that better balances reference-motion fidelity and stability for actions involving large center-of-mass changes.

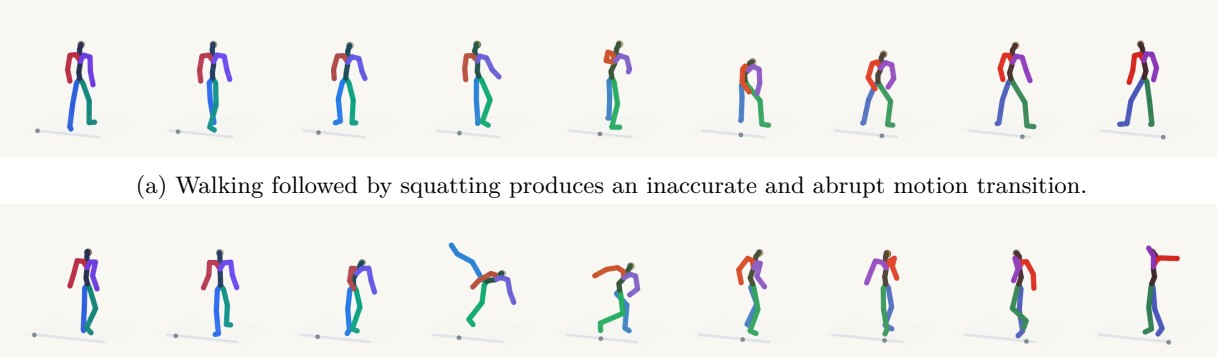

(a) Walking followed by squatting produces an inaccurate and abrupt motion transition.

(b) A standing forward jump followed by walking produces inaccurate, abrupt, and visually unnatural poses.

Figure S14: Representative PedAnimator motion-execution failures. Each sequence progresses from left to right. Actions involving large center-of-mass changes expose a trade-off in the current controller between accurate action execution and maintaining physical balance.

## S11  Limitations

The current framework has several limitations. First, ChatAni currently focuses on pedestrians and vehicles and does not model cyclists or cyclist-specific dynamics, such as balance, pedaling, bicycle steering, and cyclist-vehicle or cyclist-pedestrian interactions. Complex multi-actor cyclist scenes therefore remain outside the current scope. Second, although ChatAni accepts natural-language descriptions, the executable behaviors are constrained by the behavior categories, controller objectives, and animation primitives supported by the current system. Thus, ChatAni does not yet support fully open-ended semantic generation of arbitrary new interaction behaviors. Third, the current high-level planner relies on the proprietary GPT-4 model. We provide a limited comparison with GPT-3.5 and Llama-3 70B in Appendix S7, but have not systematically evaluated more recent backbones, a wider range of model scales, or sensitivity to prompt design. Fourth, when surrounding traffic blocks a requested vehicle maneuver, the current high-level coordination mechanism may prioritize conservative collision avoidance and produce a command-inconsistent plan, as shown in Fig. S13. Fifth, PedAnimator may prioritize physical balance over accurate action execution for motions involving large center-of-mass changes, resulting in inaccurate or visually unnatural motions, as shown in Fig. S14. Sixth, pedestrian-vehicle collision-related scenarios are represented at the level required by our evaluated cases, but fine-grained physical contact dynamics are not modeled in high fidelity. For example, detailed impact response, body deformation, contact geometry, and post-impact recovery are left for future work. Finally, the downstream augmentation experiments use a single seed and the DriveLM result is evaluated on generated hazardous scenes; these findings should therefore be interpreted as preliminary evidence under the reported settings rather than a general performance or safety guarantee.

## S12  Broader Impacts

This work serves dual purposes: (1) as a street-scene animation generator, it can support simulation platforms with diverse, physically plausible pedestrian/vehicle animations for corner-case scenarios while supporting game/film production through efficient animation synthesis; (2) as a generative prior, ChatAni-produced animations can guide video generation models for controlled outputs. Technically, ChatAni explores the integration of multi-LLM-agent role-playing in traffic scenarios and addresses multi-pedestrian physical interactions in street environments. Key components—interaction training protocols, task-masking mechanisms, and body-masked Adversarial Motion Priors—may be transferable to physics-based human animation and robotic control systems. This work demonstrates no evident potential societal negative impacts or unethical misuse scenarios.

