# OpenReview forum: "ChatAni: Language-Driven Multi-Actor Animation Generation in Street Scenes"
_TMLR — Under review for TMLR_

### Review · Reviewer_LzzV · 2026-06-11

**Summary Of Contributions:**

This paper presents ChatAni, a system for generating multi-actor street scene animation from natural language instructions. The system consists of 1) PedAnimator, a unified multi-task controller generating pedestrian animations; 2) VehAnimator, a vehicle controller that plans trajectories and generates vehicle animations; 3) a multi-agent framework that generates high level planning for all participants from user input as input of low level controller PedAnimator and VehAnimator.

Strength:
* This paper tackles an important problem in animation generation for autonomous driving simulator, where existing work fails to generate complicated interactions among pedestrians and vehicles. This paper is a systematic work that combine high-level and low-level planning and utilizes the advance of intelligence in LLMs.
* The paper is detailed and easy to follow.
* PedAnimator utilizes a novel interaction training pipeline and trains a unified policy that combines trajectory following and motion imitation.

Weakness:
* The evaluation of the system is somehow qualitative. Sec 4.2 discusses a few scenarios and provide sample animations in figures, but lacks quantitive evaluation.

**Audience:**

Yes

**Audience Explanation:**

This paper pedestrian-vehicle animation generation for autonomous driving simulation, which is a specialized field. But general ML methods are adopted, such as unified policy training, multi agent design, therefore this paper is interesting to some audience.

**Broader Impact Concerns:**

No concerns.

**Claims And Evidence:**

Yes

**Claims Explanation:**

The paper provides evaluation of each component in Sec 4, which show clear improvement over baseline. It also provides sample animations in figures showing empirically better results than previous work.

**Requested Changes:**

* Page 8 has some layout issue of figures. Figure 6 is missing, while the figure on the bottom of page 8 is partially shown and has no title.
* See weakness. Clarify the best practice for evaluating the system, and provide additional evaluation results if needed.

---

> ### Author Response · Authors · 2026-07-19
> **Response to Reviewer LzzV**
>
> We thank the reviewer for the positive assessment of ChatAni and for highlighting the importance of combining high-level language-based planning with low-level pedestrian and vehicle animation. We also appreciate the two concrete suggestions concerning the **figure layout** and the need for a stronger **full-system quantitative evaluation**. In the revision, we corrected the reported layout issue and added a structured end-to-end evaluation of the complete ChatAni pipeline. **All corresponding changes have been incorporated into the revised manuscript and can be inspected directly in Sec. 4.2, Table 2, and the supplementary evaluation protocol.**
>
> ### 1. Figure Layout
>
> Thank you for pointing out the layout problem. The affected large-scale crowd-scene figure was previously placed using a wrapped-figure environment, which could cause the figure to extend beyond the available page area and become separated from its caption.
>
> We replaced it with a **standard floating figure** and adjusted its size and placement. Figure 6 is now fully displayed together with its corresponding caption, and the following figure is no longer partially clipped at the bottom of the page. We also visually inspected the surrounding pages after recompilation to confirm that the figures remain within the margins and are not separated from their titles or captions.
>
> ### 2. Best Practice and Additional Full-System Evaluation
>
> We agree that the original Sec. 4.2 relied too heavily on representative qualitative examples.
>
> For a language-driven heterogeneous animation system, we adopt a **two-level evaluation protocol** that combines complementary component-level and full-system evidence:
>
> - **Component-level quantitative evaluation**, using automatic metrics for trajectory tracking, motion imitation, interaction completion, vehicle control, and high-level command matching.
> - **Full-system structured human evaluation**, assessing properties that cannot be reliably captured by a single automatic metric, such as whether all requested actors are present, whether the natural-language command is satisfied, whether the intended interaction occurs, and whether the complete scene is semantically valid.
>
> Following this evaluation strategy, we added a new **end-to-end full-system evaluation** containing **120 generated scenes** across six prompt categories: **single-agent, vehicle–vehicle, pedestrian–vehicle, pedestrian–pedestrian, compound, and crowd/abstract prompts**, with 20 scenes in each category.
>
> We recruited **20 participants**, each of whom evaluated 30 randomly assigned prompt-result pairs balanced across the six categories. Each generated scene was therefore evaluated by **five different participants**, resulting in **600 prompt-result evaluations**. For each pair, participants provided five binary judgments:
>
> - **Actor coverage**
> - **Command matching**
> - **Interaction success**
> - **Map validity**
> - **Overall scene success**
>
> This produced **3,000 binary judgments** in total. We report positive-rating percentages together with **Wilson 95% confidence intervals**.
>
> Across all evaluated scenes, ChatAni obtains **88.5% actor coverage**, **82.7% command matching**, **78.8% interaction success**, **84.2% map validity**, and **73.8% overall scene success**. The results also show lower performance on **compound** and **crowd/abstract** prompts, which we now discuss as more challenging full-system cases rather than reporting only successful examples.
>
> This new evaluation complements the existing component-level metrics and qualitative animations with a broader quantitative assessment of the complete ChatAni pipeline. The complete results are reported in the revised **Sec. 4.2 and Table 2**, while participant assignment, evaluation criteria, and statistical details are provided in the supplementary evaluation protocol.

---

### Review · Reviewer_WnBM · 2026-06-17

**Summary Of Contributions:**

The authors propose ChatAni which can generate animations involving vehicles and pedestrians. It is an interesting idea to generate animations using text.

**Additional Comments:**

This paper requires much more editing before it is publication-ready.

**Audience:**

No

**Audience Explanation:**

This article's presentation is incoherent with no organization to its narrative. The overall idea might be interesting, but there appears to be no reproducibility here. I also am skeptical of the evaluation results, as they are poorly explained. Additionally, Figure 9 for instance, the bar plots do not start from zero which deceptively shows a "larger difference" between the three comparisons, even though they are likely not substantially different. Figure 10 has an x-axis measured in percentages, but the actual bar plot values are shown in range [0, 1], so they don't align with the x-axis.

**Claims And Evidence:**

No

**Claims Explanation:**

Figure 1, caption: "ChatAni produces physically plausible and realistic motions across diverse interaction modalities and heterogeneous actors" ---> how do you show that the results are plausible or realistic when LLM generated content frequently creates people having hands with incorrect number of fingers or performing motions that are impossible? Also, how come Dosovitskiy et al. (2017) page 1 paragraph 2's work is described as "especially not realistic" when generating from templates whereas ChatAni is generating scenes from a LLM?

Figure 1 looks like it is AI-generated. The Street Animation (bottom right) looks strange; the stacking should be more obvious this is a sequence of images.

The introduction is quite unpolished - diving directly into what is being introduced without much set up for the problems that are to be solved, with lots of verbiage that carries little meaning.

In Related Work, "Human animation generation", this section does a very poor attempt at summarizing or explaining these works. A set of works are all cited and then all is said about them is "introduce further control conditions. However, these approaches do not account for physical constraints." I am unsure what to make of this description. Further elaboration is needed. Additionally, "Our PedAnimator considers the interaction behaviors and is trained as a unified policy for multiple scenarios also including following, imitation." The last part of this sentence is not grammatically correct and makes little sense.

Related Work, "Vehicle traffic generation", again this feels like an insufficient coverage of cited works. You should elaborate a bit more about unconditioned generation. Are they really unconditioned? I doubt this. There must be some form of conditions provided to these generations, even if simple. The sentence "And these language-controlled methods cannot achieve precise control over different vehicles or other participants." is abrupt and sudden where I am unsure what to interpret from it.

Add a space in "Large language models and agents" between "fine-tuning them" and "Hu et al. (2021)".

The writing style of the methodology section causes me to doubt this work's reproducibility. No explicit mention of which physics engine or LLM is used. Poor description of state space - could use a diagram to illustrate how these joint positions relate to an animation. "trajectory following and motion imitation" <--- page 5, paragraph 1, what are we following and imitating if this is being generated?

"To enable specific parts during training or testing, we introduce a task masking mechanism." That is very random and out of place. Care to elaborate? I am not able to follow this writing.

This whole methodology section screams a need for a diagram to illustrate how each of these proposed items such as states & multi-task unified training, action hierarchical control, reward design, etc. interact with each other. Methodologies are randomly name-dropped with little or poor explanation as to what they do - PULSE by Luo et al. 2024 - why are we using it? The motivation to use this is barely existent, only mentioning "... but such spaces lack inherent priors, often leading to locally unrealistic actions for completing specific tasks." I understand the last part, but what priors are we discussing??

AMP with body mask and warm-up training: Random mention again of "following or imitation tasks" which have not been properly introduced yet.

Figure 3 & 4, why are there fire and snow emojis in a research article?

Page 7, "We employ the bicycle model to model the vehicle dynamic transition process." What bicycle model? Can you cite it?

Significant issues continue from here, that are lengthy beyond what I will write in a review.

**Requested Changes:**

"S10 Border impacts" should be "S10 Broader Impacts"

Page 8, bottom right, an image is outside margins.

Page 9, top right, large white space of potentially missing item.

"S3 Ethic, Reproducilibity Statement and Usage of LLM" should be "S3 Ethic, Reproducibility Statement, and Usage of LLM"; additionally, "The LLM in this work is used for writing enhancement purposes." I appreciate you disclosing the use of LLMs, but it is quite apparent LLMs were used in preparation of this article, which is actually harming the communication of your ideas. It makes the writing style overly verbose while saying nothing. After reading 6 pages, all I am left with understanding is you have a ChatAni and it interacts with PedAnimator and VehAnimator. I don't really understand on a deep sufficient level how any of this is done, which completely detracts from the essence of reproducible research. The explanations are way too hand wavy.

---

> ### Author Response · Authors · 2026-07-19
> **Response to Reviewer WnBM(1)**
>
> ## Response to Reviewer WnBM
>
> We thank the reviewer for the detailed feedback. Before addressing the individual comments, we would like to clarify an important point about the original submission. **Several elements described in the review as absent were already included in the manuscript:**
>
> - The original paper contained **three methodology diagrams**: an overall ChatAni/Multi-LLM-Agent framework diagram (Fig. 2), a detailed PedAnimator training-and-inference diagram (Fig. 3), and a VehAnimator diagram (Fig. 4). In particular, the original Fig. 3 already showed the task states, task mask, policy, latent action decoder, physics simulation, AMP discriminator, task reward, and PPO feedback in one connected pipeline.
> - The original **Implementation Details** explicitly identified **Isaac Gym, the SMPL humanoid, GPT-4, and the ChatSim rendering pipeline**.
> - The original method and appendix mathematically defined the **trajectory, reference-motion, interaction-target, and proprioceptive states**, as well as task masking, trajectory-following and motion-imitation rewards, PULSE-based latent action control, and body-masked AMP.
> - The original VehAnimator subsection provided the **bicycle-model state-transition equations and variable definitions** immediately after introducing the model.
> - The original experiments were not solely qualitative: Sec. 4 reported quantitative evaluations of **PedAnimator motion quality, imitation, following, and interaction; VehAnimator position/velocity tracking; high-level command compliance and map validity; user preference; ablations; and downstream prediction and DriveLM results**.
>
> Thus, the central issue was not that these components or diagrams were missing. Rather, some information was distributed across the method, experiments, and appendix, and several captions and transitions relied too heavily on terminology and visual conventions familiar to the physics-based animation, character-control, and vehicle-control communities. We recognize that a TMLR paper should also be self-contained for readers outside these immediate areas. We therefore made the existing material easier to locate and interpret by adding explicit definitions, motivations, cross-references, citations, legends, and implementation details, and by revising the existing figures and captions.
>
> More broadly, the revision makes substantial changes in four areas: **(1) clarifying the role of the LLM and the scope of our claims; (2) restructuring the introduction and related work; (3) expanding and reorganizing the methodology while revising the existing framework diagrams; and (4) correcting the evaluation figures, formatting issues, and reproducibility statement.** **All changes described below have been incorporated into the revised manuscript and can be inspected directly in the updated introduction, related work, methodology, experiments, figure captions, and supplementary material.**
>
> ### 1. Figure 1 claims, LLM-generated content, and the CARLA comparison
>
> We first clarify a distinction that was already part of the original system design: **ChatAni does not use an LLM or image-generation model to generate pixels, human appearances, body meshes, or anatomical details such as fingers.** The original method separated the Multi-LLM-Agent planner from PedAnimator and VehAnimator: the LLM agents produce **structured high-level plans**, including participant roles, behavior descriptions, schedules, interaction groups, and trajectory keypoints, while the low-level animators control simulated humanoid joints and vehicle states. The concern about incorrect numbers of fingers therefore pertains to pixel-level image generation and is not applicable to the generation mechanism evaluated in this work.
>
> We nevertheless agree that the original Figure 1 caption did not restate this separation clearly enough and that its broad wording could invite the wrong interpretation. We revised the caption and introduction to explicitly distinguish **language-based planning** from **low-level animation execution**, and we replaced broad claims such as “realistic” and “precise” with scoped descriptions tied to the evaluated properties, including **physically plausible pedestrian motion**, **language-guided control**, and **kinematically compliant vehicle animation in the evaluated settings**.
>
> We also revised the comparison with CARLA. Our intention was not to describe CARLA as inherently unrealistic. CARLA provides effective manually authored assets and behavior templates. Our concern is that repeatedly applying a limited collection of templates can result in **rigid or repetitive behavior**, while scaling such templates to language-conditioned, fine-grained multi-actor interactions requires substantial manual design. The revised manuscript now makes this distinction explicit.

---

> > ### Author Response · Authors · 2026-07-19
> > **Response to Reviewer WnBM(2)**
> >
> > ### 2. Figure 1 construction and temporal presentation
> >
> > We clarify that Figure 1 was **manually assembled by the authors** using schematic elements and outputs from our animation/rendering pipeline; it was not produced by a text-to-image model. To make its provenance directly inspectable, the **original editable PPT file for Figure 1 is included with the submitted supplementary materials**. The planning, map, pedestrian-animation, vehicle-animation, and rendered-sequence panels are drawn from outputs of our actual pipeline and correspond to qualitative examples, component results, and animation sequences reported in the experiments and supplementary video. The editable source and these correspondences provide additional evidence that the figure was manually constructed from experimental outputs rather than generated as a synthetic illustration.
> >
> > We agree that the previous Street Animation panel did not clearly communicate temporal progression. We revised the figure by adding **explicit \(t_1\), \(t_2\), and \(t_3\) labels and a time-direction arrow**, and the revised caption states that these are consecutive frames from the same generated animation.
> >
> > ### 3. Introduction and problem formulation
> >
> > We substantially revised the introduction to establish the problem before presenting ChatAni. The revised introduction first defines street-scene animation and explains its relevance to driving simulation, game and film production, and controllable video generation.
> >
> > It then identifies two concrete technical challenges:
> >
> > - **Scene-level planning:** natural-language commands must be converted into map-constrained, temporally consistent, multi-actor plans.
> > - **Low-level execution:** these plans must be executed without unstable human motion, invalid contacts, or kinematically inconsistent vehicle states.
> >
> > We also removed or replaced vague and repetitive phrases with concrete descriptions of **pedestrian-vehicle interactions, map constraints, actor-level control, trajectory planning, and executable low-level animation**.
> >
> > ### 4. Related work
> >
> > We agree that the original related-work section grouped too many papers together without adequately explaining their contributions or differences.
> >
> > For **human animation generation**, we now distinguish among:
> >
> > - kinematic motion-generation methods;
> > - controllable kinematic methods using text, trajectory, joint, or timing conditions; and
> > - physics-based character-control methods.
> >
> > We clarify that kinematic methods generally output pose sequences without explicitly enforcing balance, contacts, external perturbations, or physically reactive interactions. We also describe the focus of prior physics-based methods on locomotion, predefined skills, skill composition, and pedestrian control. The unclear sentence about PedAnimator has been replaced with a precise description of its unified support for **trajectory following, motion imitation, and selected interaction tasks**.
> >
> > For **vehicle traffic generation**, we replaced the misleading term “unconditioned generation” with **“scene-context-conditioned or weakly controlled generation.”** We now explain that these methods may condition on maps, histories, initial states, or sampled goals, while our setting additionally requires language-conditioned actor-level control and low-level pedestrian/vehicle animation. We also clarify the respective compatibility boundaries of LCTGen, CTG++, Talk2Traffic, and other baselines.
> >
> > ### 5. Citation spacing
> >
> > We corrected the missing space between “fine-tuning them” and the citation to Hu et al. (2021).

---

> > > ### Author Response · Authors · 2026-07-19
> > > **Response to Reviewer WnBM(3)**
> > >
> > > ### 6. Implementation details, state space, and conditioning signals
> > >
> > > We respectfully clarify that the implementation choices identified in the review were **explicitly stated in the original submission**, although they appeared in the Implementation Details subsection rather than at their first point of use in the method. That subsection stated that PedAnimator uses **Isaac Gym** with an **SMPL humanoid**, that the LLM components use the **GPT-4 API**, and that final rendering follows the **ChatSim pipeline**. The appendix additionally identified **Blender Cycles** as the renderer. We agree that requiring readers to recover these choices from later sections made the method harder to follow, so the revised method now repeats them at the relevant locations and adds direct cross-references. It now makes immediately visible that:
> > >
> > > - **PedAnimator is trained in Isaac Gym** using an **SMPL humanoid**;
> > > - **GPT-4** is used by the Multi-LLM-Agent planner;
> > > - **VehAnimator uses a bicycle-model transition**; and
> > > - rendering follows the **ChatSim/Blender pipeline**.
> > >
> > > The state space was likewise not absent from the original submission. The main method defined \(\mathcal{S}^{p}_{traj}\), \(\mathcal{S}^{p}_{mo}\), \(\mathcal{S}^{p}_{tar}\), and \(\mathcal{S}^{p}_{prop}\), including the joint position, rotation, velocity, and angular-velocity components; the appendix further specified the 24-joint dimensions and normalization. This representation follows established physics-based character controllers such as **Pacer, Pacer+, and PHC** and is not presented as a separate contribution of ChatAni. To improve readability, the revision now additionally explains in plain language that these quantities are **policy observations**, not rendered pixels. At each simulation step, the policy observes these states, produces a latent control action, and the physics simulator updates the humanoid state to generate the animation sequence.
> > >
> > > The original state definitions also specified what is followed and imitated: \(\mathcal{S}^{p}_{traj}\) contains future steps of the planned trajectory, while \(\mathcal{S}^{p}_{mo}\) contains the joint states of the reference motion. **Trajectory following** and **motion imitation** are standard task formulations in physics-based character animation and are used with the same general meaning in Pacer/Pacer+ and related humanoid-control work. However, we agree that the original text expected too much familiarity with this terminology. We now define the tasks before using their names: trajectory following tracks a target root path from the planner or reference data, while motion imitation tracks a reference motion signal, such as an upper-body motion clip. PedAnimator therefore conditionally generates a physically simulated full-body sequence under trajectory and/or reference-motion constraints; “generation” does not mean that it is unconditioned.
> > >
> > > We additionally clarify the planner interface: the Oracle and Actor agents produce **JSON-formatted structured outputs**, with selected text fields containing natural-language behavior descriptions. Deterministic follow-up functions convert these outputs into the state, schedule, trajectory, and behavior structures consumed by the low-level animators.
> > >
> > > ### 7. Task masking
> > >
> > > The original submission did describe the task mask in both the method and appendix: it stated that the task-relevant state channels are unmasked for each episode, unused channels are zero-masked, and the corresponding rewards are activated during training or inference. The issue was therefore not an absent mechanism, but the unclear introductory phrase “enable specific parts” and insufficient motivation before the definition. Masking inactive conditioning channels is a common implementation pattern in unified or multi-task policies, but **our particular task-masking design and its role in PedAnimator still require an explicit explanation**. We rewrote this transition and now explain why task masking is required before describing how it operates.
> > >
> > > PedAnimator uses a single policy for several task types, but different episodes require different observations and rewards. For example, trajectory following uses the planned trajectory state but does not require an interaction target, while an interaction task may use both trajectory and target states.
> > >
> > > The **task mask activates the observations and reward terms relevant to the current episode and masks unused task-specific inputs**. We now describe this behavior separately for training and inference and avoid terminology that could be confused with body-part masking.

---

> > > > ### Author Response · Authors · 2026-07-19
> > > > **Response to Reviewer WnBM(4)**
> > > >
> > > > ### 8. Methodology diagrams and the role of PULSE
> > > >
> > > > We respectfully clarify that the original submission **already contained the requested methodology diagrams**. Fig. 2 presented the overall Multi-LLM-Agent planning framework, Fig. 3 presented PedAnimator, and Fig. 4 presented VehAnimator. Moreover, the original PedAnimator diagram already connected the task states, on-demand task mask, policy network, latent action decoder, physics simulation, task reward, AMP discriminator, and PPO update, with separate training and inference regions. The original VehAnimator diagram likewise showed the current state, policy, control actions, vehicle transition, next state, tracking/smoothness rewards, and learning loop. Thus, these diagrams were not introduced for the first time in response to the review.
> > > >
> > > > We do recognize that their legends, captions, arrows, and surrounding prose did not explain every relationship clearly enough for a reader outside the immediate field. We therefore revised the **existing** PedAnimator and VehAnimator diagrams and expanded their captions. The updated PedAnimator figure makes the following flow especially explicit:
> > > >
> > > > **task states → task mask → policy network → latent action → action decoder → physics simulation → task and AMP rewards.**
> > > >
> > > > The revised text also describes this process as a closed-loop controller rather than introducing the components independently.
> > > >
> > > > The motivation and mechanism for PULSE were also present in the original method: it explained that direct per-DoF PD control lacks a learned motion prior, and that the policy instead outputs to PULSE's pretrained latent space, whose decoder maps the latent feature to joint-control signals drawn toward the pretraining distribution. **PULSE is an existing module rather than a contribution of ChatAni**, and pretrained latent action priors are an established design in physics-based character control. We nevertheless expanded this explanation and now state more directly that the frozen PULSE decoder converts a compact latent action into coordinated joint-control signals, reducing the action-search space and biasing the controller toward action patterns learned from human motion data. ChatAni does not claim the PULSE latent space itself as a contribution.
> > > >
> > > > Finally, we distinguish the roles of the rewards: **task rewards specify what the character should accomplish**, while **AMP rewards regularize how the resulting motion should look**.
> > > >
> > > > ### 9. AMP, body masking, and warm-up training
> > > >
> > > > We respectfully clarify that “following” and “imitation” were introduced earlier in the same PedAnimator subsection, where the original paper defined the trajectory state and reference-motion state; the appendix also gave their reward equations. They were therefore not new tasks introduced in the AMP paragraph. However, the original paragraph should have referred back to those definitions more explicitly. We revised it to connect “following” and “imitation” directly to the previously defined trajectory-conditioned and reference-motion-conditioned tasks. **AMP is a standard adversarial motion-prior framework in physics-based character animation**, but the way its reference data interacts with our following, imitation, and interaction objectives is specific to our training pipeline and therefore also required a clearer explanation.
> > > >
> > > > Reference clips from these tasks provide useful locomotion and upper-body priors, but they do not necessarily contain the contact patterns required by interaction tasks such as pushing or patting. Directly applying these clips as AMP references can therefore encourage motion that looks plausible in isolation but fails to satisfy the interaction objective.
> > > >
> > > > This motivates our two-stage training strategy: an initial **AMP-free warm-up phase** first establishes task completion, after which AMP regularization improves motion plausibility. The **body-masked AMP** additionally excludes interaction-critical joints from selected discriminator calculations so that the controller retains the flexibility required for contact execution.
> > > >
> > > > ### 10. Trainable and frozen notation in Figures 3 and 4
> > > >
> > > > The fire and snowflake symbols are a commonly used compact convention in machine-learning architecture diagrams to indicate **trainable** and **frozen** modules; they were not decorative emojis. The original figures used them with this standard meaning. We nevertheless agree that relying on the convention without an explicit legend could be unclear to readers outside the immediate animation/control community. The revised figures now contain explicit **Trainable** and **Frozen** legends, and the captions define the notation and distinguish the training and inference paths.
> > > >
> > > > In the PedAnimator figure, the fire symbol denotes trainable policy/discriminator modules and the snowflake denotes the frozen action decoder. In the VehAnimator figure, only the trainable marker is shown because no frozen module is depicted in that diagram.

---

> > > > > ### Author Response · Authors · 2026-07-19
> > > > > **Response to Reviewer WnBM(5)**
> > > > >
> > > > > ### 11. Bicycle model
> > > > >
> > > > > The bicycle model is a standard kinematic abstraction in vehicle-control and autonomous-driving research. More importantly, the original manuscript did not merely name it: **immediately after the cited sentence, it provided the complete transition equations for \(\dot{x}\), \(\dot{y}\), \(\dot{\phi}\), and the slip angle \(\beta\), and defined the associated state variables and vehicle parameters.** The missing element was a literature citation, not the mathematical specification of the model. We have now added a citation to Rajamani’s *Vehicle Dynamics and Control* and polished the surrounding definition. Additional parameter settings remain available in the appendix. We also consistently describe VehAnimator as **kinematics-based** or **kinematically compliant**, rather than implying that it is a general-purpose physics simulator.
> > > > >
> > > > > ### 12. Figures 9 and 10
> > > > >
> > > > > We agree that the previous visualization could exaggerate differences because Figure 9 used truncated axes without clearly indicating them. Figure 9 now uses **explicitly marked broken y-axes**, and the exact values are annotated on every bar.
> > > > >
> > > > > We also corrected Figure 10 so that the axis and bar labels use the same percentage representation. The three displayed collision rates are now consistently reported as **8.6%**, **75.3%**, and **17.3%**. The accompanying text also reports the corresponding Wilson 95% confidence intervals and clarifies that the experiment evaluates generated hazardous scenes rather than general driving safety.
> > > > >
> > > > > ### 13. Evaluation scope, protocols, and uncertainty
> > > > >
> > > > > We also respectfully clarify that the original Sec. 4 already contained extensive quantitative component-level evaluation. PedAnimator was evaluated using **FID, diversity, low-speed FID/diversity, MPJPE, trajectory-following error, interaction success, user preference, and ablations**. VehAnimator was evaluated using **position and velocity tracking errors, user preference, and ablations**. The high-level planner was evaluated using **command matching, within-road validity, and user preference**, and the paper additionally reported downstream prediction and DriveLM results. The qualitative figures and supplementary video illustrated these quantitative experiments; they were not the sole evidence for the method.
> > > > >
> > > > > We agree, however, that the original paper did not consolidate the protocols and uncertainty estimates clearly enough, and that its complete-system evaluation was weaker than its component evaluation. The revision therefore adds:
> > > > >
> > > > > - a new **120-scene end-to-end evaluation** across six prompt categories, with five independent ratings per scene, 600 prompt-result evaluations, and 3,000 binary judgments;
> > > > > - explicit full-system criteria for **actor coverage, command matching, interaction success, map validity, and overall scene success**;
> > > > > - **Wilson 95% confidence intervals** for binary and preference metrics;
> > > > > - **mean ± standard deviation over three independent runs** for continuous PedAnimator and VehAnimator metrics;
> > > > > - explicit denominators of **1,000 episodes per PedAnimator interaction cell** and **300 trials per DriveLM condition**; and
> > > > > - a consolidated appendix table specifying participant counts, items per participant, total judgments or votes, anonymization, randomized display order, and the calculation of preference rates.
> > > > >
> > > > > These additions make the scope of each result and the distinction between component-level and complete-system evidence explicit. They also expose the more difficult compound and crowd/abstract cases rather than relying only on favorable examples.

---

> > > > > > ### Author Response · Authors · 2026-07-19
> > > > > > **Response to Reviewer WnBM(6)**
> > > > > >
> > > > > > ### 14. Typographical, layout, LLM-usage, and reproducibility issues
> > > > > >
> > > > > > We corrected **“Border Impacts”** to **“Broader Impacts”** and changed the appendix heading to **“Ethics, Reproducibility Statement, and Usage of LLMs.”**
> > > > > >
> > > > > > We also revised the relevant figure placement. In particular, the large crowd-scene visualization is now placed as a standard float rather than a wrapped figure, preventing it from extending beyond the page margin or producing unintended blank regions. We visually inspected the revised pages after compilation.
> > > > > >
> > > > > > The LLM-usage statement now explicitly distinguishes two uses:
> > > > > >
> > > > > > - **GPT-4 is a core high-level planning component** used by the Oracle and Actor agents to interpret instructions and generate structured plans.
> > > > > > - LLM-based tools may separately have been used for minor language polishing.
> > > > > >
> > > > > > The statement also clarifies that GPT-4 is not used to train PedAnimator or VehAnimator and is not used to evaluate the generated results.
> > > > > >
> > > > > > Finally, the original submission already provided substantial reproducibility material, including method equations, state and reward definitions, network and training settings, pseudocode, prompts, and selected code. We agree, however, that the previous statement did not inventory these artifacts precisely and that it overstated the completeness of the package. The revised statement now accurately lists the supplied materials: **training code and configuration files for the main experiments, selected evaluation code, high-level planning implementation and design, prompt templates, model settings, pseudocode, and evaluation protocols**. We also synchronized the methodology and pseudocode so that the PedAnimator reward weighting, Isaac Gym pedestrian transition, and bicycle-model vehicle transition are described consistently. Because the limited revision period was insufficient to clean, validate, document, and package the entire research codebase, generated data, and trained checkpoints to a public-release standard, these remaining artifacts are not yet included. The current materials nevertheless provide concrete support for inspecting and reproducing the main implementation and evaluation procedures. After completing the required curation, we will release the **full codebase, generated data, and trained checkpoints**, subject to storage constraints and third-party dataset licenses; until then, we do not describe the current artifact package as fully complete.

---

### Review · Reviewer_dcqR · 2026-07-06

**Summary Of Contributions:**

1. This works look at generating interactive and controllable multi-actor animations based on language instructions.

2. They do this using animator policies for pedestrians and vehicles.

3. They use multi-agent system to simulate trajectories of participants, this is similar to works which treat LLMs as world models and then make use of them in simulations.

**Audience:**

Yes

**Audience Explanation:**

This is definitely of interest to the community as it enables better simulations for autonomous driving especially with adversarial actors or underrepresented pedestrian motions.

**Claims And Evidence:**

No

**Claims Explanation:**

What is supported by claims: PedAnimator is evaluated with FID/diversity, imitation error, following error, interaction success rates, user preference, and ablations; VehAnimator is evaluated with position/velocity tracking error and ablations; and the LLM planning module is compared against LCTGen and ChatSim on command matching, within-road rate, and user preference.

However, several major claims are broader than the evidence justifies. The paper repeatedly claims that ChatAni is the “first” system for language-driven multi-actor street animation and that it generates "interactive, realistic, and controllable" scenes, but the evidence is mostly a mix of component-level tests, selected qualitative examples, and small or underspecified user studies.

The full-system evaluation is relatively weak: the appendix reports only 20 final results judged by 10 participants, with 69% agreement that outputs were realistic and command-aligned, which is promising but not enough to strongly validate the broad system-level claims.

The downstream application evidence is also underdeveloped. Figure 9 suggests that ChatAni augmentation improves prediction metrics, and Figure 10 suggests reduced DriveLM collision rates after fine-tuning on ChatAni-generated hazards. But the paper provides limited detail about variance, statistical significance, data splits, possible leakage, number of random seeds, and comparisons to other augmentation approaches. The claim that ChatAni "significantly enhances safety" is therefore too strong as written.


There are also clarity and accuracy concerns. The paper says the LLM is used for writing enhancement in the ethics/reproducibility statement, but the method itself relies on GPT-4 (there are substantially better open source models to test the claims) as a core planning component. The user-study methodology is scattered and somewhat inconsistent across sections, with different participant counts and sample counts appearing in the main text and appendix.

**Requested Changes:**

Requested changes in simple language:

1. **Tone down the big claims** unless stronger proof is added. Claims like first system, realistic, precise control, and significantly enhances safety are stronger than the current evidence supports.

2. **Add a stronger full-system evaluation.** The paper mostly evaluates separate parts. The complete ChatAni system should be tested on many more prompts, scenes, and interaction types, not just selected examples.

3. **Report failures, not only successes.** Show cases where ChatAni fails: bad language parsing, unrealistic motion, collisions, wrong timing, poor interaction, or unstable pedestrians/vehicles.

4. **Clarify the user studies.** The paper mentions different user-study sizes: 100 users, 1000 users, and 10 participants in different places. Explain exactly who participated, how many samples they saw, whether results were randomized/blinded, and how preferences were calculated.

5. **Add confidence intervals or statistical tests.** Tables report improvements, but there are no error bars, variance, or significance tests. Add these for the main metrics.

6. **Make downstream application claims more careful.** The prediction and DriveLM results are promising, but the paper should provide more details on seeds, data splits, leakage checks, and comparisons before claiming safety improvement.

7. **Fix the LLM usage statement.** The appendix says the LLM is used for writing enhancement, but the method uses GPT-4 as a core planning component. This should be corrected clearly.

8. **Improve reproducibility.** Provide full code, data, checkpoints, prompts, configs, evaluation scripts, and model settings, or avoid saying the work is fully reproducible until they are available.

9. **Add stronger baseline comparisons.** Compare against the closest possible methods for each claim, and explain clearly when a baseline cannot handle pedestrians, language, or interactions.

10. **Clarify limitations.** The paper should be more direct about what ChatAni cannot yet do, especially complex cyclist scenes and fine-grained pedestrian-vehicle physical collisions, which are listed as future work.

---

> ### Author Response · Authors · 2026-07-19
> **Response to Reviewer dcqR(1)**
>
> ## Response to Reviewer dcqR
>
> We sincerely thank the reviewer for the detailed and constructive assessment. We agree that the original manuscript did not sufficiently distinguish **component-level evidence**, **full-system evidence**, and **preliminary downstream results**, and that several claims were broader than the supporting evaluation. **All corresponding changes described below have been incorporated into the revised manuscript and can be inspected directly in the updated abstract, introduction, methodology, experiments, conclusion, and supplementary sections.** In addition to answering each point individually, we revised the relevant surrounding text so that these clarifications are reflected consistently throughout the paper rather than appearing only in this response.
>
> The revision makes four main groups of changes:
>
> - **Stronger and more balanced system-level evidence:** we add an end-to-end evaluation of **120 complete ChatAni generations**, covering six prompt categories, together with four representative failure cases.
> - **Clearer evaluation and uncertainty reporting:** we consolidate all user-study protocols, correct participant and response counts, add **Wilson 95% confidence intervals** for the main binary and preference-based metrics, and report continuous PedAnimator and VehAnimator metrics as **mean ± standard deviation over three independent runs**.
> - **More careful downstream and reproducibility claims:** we document seeds, splits, augmentation budgets, leakage checks, DriveLM settings, and available artifacts, while explicitly identifying the downstream results as **single-seed preliminary evidence**.
> - **Better-scoped comparisons and limitations:** we add **UniPhys** and **Talk2Traffic**, explain baseline compatibility boundaries, disclose the reliance on **GPT-4**, and expand the limitations concerning cyclists, open-ended behaviors, planning failures, PedAnimator failures, contact physics, and downstream generalization.
>
> Our detailed responses follow.
>
> ### 1. Toning Down Broad Claims
>
> We agree that claims such as **“the first system,” “realistic,” “precise control,”** and **“significantly enhances safety”** were stronger than the original evidence justified.
>
> We revised these claims throughout the **abstract, introduction, method, experiment captions, downstream application section, and conclusion**. In particular:
>
> - Absolute novelty claims such as **“the first system”** are replaced with the scoped description of ChatAni as **a language-driven framework for multi-actor street-scene animation**.
> - “Realistic” is replaced where appropriate by more measurable descriptions such as **physically plausible**, **visually plausible**, or **human-like under the evaluated metrics**.
> - “Precise control” is replaced by **command following**, **command alignment**, or **improved control under the evaluated benchmark**.
> - We no longer claim that ChatAni augmentation **significantly enhances safety**. The revised paper states only that ChatAni-augmented fine-tuning reduces collision rates in the reported generated hazardous-scene evaluation and does not constitute a general safety guarantee.
>
> These changes align the claims more directly with the reported evidence.
>
> ### 2. Stronger End-to-End Full-System Evaluation
>
> We agree that the original submission relied too heavily on component evaluations and selected qualitative examples. We therefore add a new end-to-end human evaluation covering **120 complete ChatAni generations**.
>
> The evaluation contains six prompt categories:
>
> - **single-agent**
> - **vehicle–vehicle**
> - **pedestrian–vehicle**
> - **pedestrian–pedestrian**
> - **compound**
> - **crowd/abstract**
>
> Each category contains **20 generated scenes**. We recruit **20 participants**, each of whom evaluates **30 randomly assigned prompt-result pairs**, balanced across the six categories. Each scene is therefore evaluated by five different participants, producing **600 prompt-result evaluations**. Because every evaluation contains five binary questions, the study collects **3,000 binary judgments** in total.
>
> The five criteria are **actor coverage, command matching, interaction success, map validity,** and **overall scene success**. Across all generations, ChatAni obtains:
>
> - **88.5%** actor coverage
> - **82.7%** command matching
> - **78.8%** interaction success
> - **84.2%** map validity
> - **73.8%** overall scene success
>
> We report Wilson 95% confidence intervals for every category and overall metric. Performance decreases on **compound** and **crowd/abstract** prompts, and we now discuss these categories as more challenging full-system settings rather than presenting only successful examples.

---

> > ### Author Response · Authors · 2026-07-19
> > **Response to Reviewer dcqR(2)**
> >
> > ### 3. Reporting Failure Cases
> >
> > We agree that reporting only successful examples provides an incomplete characterization of the system. We add a new **“Failure Cases and Analysis”** section containing four representative unsuccessful outputs from two observed failure modes.
> >
> > For high-level vehicle planning, we report:
> >
> > - A command requesting acceleration and a **left-lane overtake**, where surrounding vehicles block the requested maneuver and the planner instead overtakes from the right.
> > - A command requesting a vehicle to **pull over**, where surrounding traffic obstructs the path and the target vehicle only decelerates without completing the maneuver.
> >
> > These cases show that the requested behavior may be identified correctly while the current coordination mechanism fails to reconcile **exact command following, multi-vehicle coordination,** and **conservative collision avoidance**.
> >
> > We also report two **PedAnimator-only** failures:
> >
> > - walking followed by squatting;
> > - a standing forward jump followed by walking.
> >
> > These instructions do not pass through the Multi-LLM-Agent planner. MoMask first produces the reference motion, which is then executed by PedAnimator under physical constraints. Both actions require large and rapid center-of-mass changes. When reference-motion fidelity conflicts with stability, the current controller tends to prioritize **physical balance**, resulting in inaccurate execution, abrupt transitions, and visually unnatural poses.
> >
> > These examples are also incorporated into the revised limitations discussion.
> >
> > ### 4. Clarifying All User Studies
> >
> > We add a dedicated **“User Study Details”** section and a summary table that reports, for every study:
> >
> > - the number of unique participants;
> > - the number of items evaluated by each participant;
> > - the total number of evaluations, judgments, or votes;
> > - the randomization and anonymization protocol;
> > - the method used to calculate preference rates.
> >
> > Throughout the revision, **participants** denotes unique human evaluators, while **evaluations, judgments, responses,** and **votes** denote collected annotations.
> >
> > For preference studies, method identities are hidden, outputs are assigned anonymized labels, and display order is randomized. Preference is computed as the number of valid votes received by a method divided by the total number of valid votes.
> >
> > For the high-level planning study, there are **100 unique participants**. All participants evaluate every sample in the five-map, 26-sample setup for each category, producing:
> >
> > **5 maps × 26 samples × 100 participants = 13,000 responses per table cell.**
> >
> > For the new full-system study, the revision distinguishes **600 prompt-result evaluations** from the resulting **3,000 binary judgments** across five criteria. The original 20-result, 10-participant study remains in the appendix but is now explicitly labeled as a **preliminary system study**, separate from the new end-to-end evaluation.
> >
> > ### 5. Confidence intervals and statistical analysis
> >
> > We thank the reviewer for this important suggestion. In the revised manuscript, we added **uncertainty estimates** for the main reported metrics. For the main binary and preference-based metrics reported in the paper, we provide **Wilson 95% confidence intervals**. Each PedAnimator interaction result is evaluated over **1,000 episodes per table cell**. In the high-level planning evaluation, each table entry aggregates **5 maps × 26 samples × 100 participants**, corresponding to **13,000 responses per table cell**. For the end-to-end evaluation, each overall criterion is computed from **600 prompt-result evaluations**, with **3,000 binary judgments** collected across the five criteria.
> >
> > For the DriveLM experiment, each condition contains **300 trials**. The collision rates and Wilson 95% confidence intervals are **8.6% [5.9%, 12.3%]** for the original scenes, **75.3% [70.1%, 79.8%]** for the ChatAni-edited hazardous scenes, and **17.3% [13.4%, 22.0%]** after ChatAni-augmented fine-tuning.
> >
> > For continuous PedAnimator and VehAnimator metrics, we report **mean ± standard deviation over three independent runs**. We also added a statistical-analysis paragraph in the appendix explaining these calculations and avoided unsupported paired significance tests where the required paired raw measurements are unavailable.

---

> > > ### Author Response · Authors · 2026-07-19
> > > **Response to Reviewer dcqR(3)**
> > >
> > > ### 6. Downstream evaluation and DriveLM claims
> > >
> > > We thank the reviewer for this important suggestion. We revised the downstream application section to provide a more complete **evaluation protocol** and to make the associated claims more careful. In particular, we no longer describe the DriveLM results as demonstrating a general or significant safety improvement. Instead, we present them as preliminary evidence that ChatAni-generated hazardous scenes may be useful for safety-oriented data augmentation under the evaluated setting.
> > >
> > > All downstream experiments use **seed 1111**. For Waymo/MTR, training uses 243k scenes sampled from the official training split, evaluation uses the official validation split, and synthetic augmentation is generated only from training-split scenes. For Human3.6M/HumanMAC, we follow the original HumanMAC split and preprocessing protocol and augment only the training split. The compared augmentation methods use the same augmentation budget and downstream training configuration.
> > >
> > > For DriveLM, we reserve **30 Waymo training-split scenes** as held-out downstream evaluation scenes and conduct **10 trials per scene**, resulting in **300 trials per condition**. These scenes and frames are excluded from the **3,000-frame ChatAni-augmented fine-tuning set**, and we verify that there is no scene/frame overlap. The three evaluated conditions obtain collision rates of **8.6%** for the original scenes, **75.3%** for the deliberately hazardous ChatAni-edited scenes, and **17.3%** after ChatAni-augmented fine-tuning. Fine-tuning uses one epoch, a learning rate of **1e-6**, and batch size 1.
> > >
> > > These results show a reduction in collision rate on our **generated hazardous-scene benchmark**, but they should not be interpreted as a general safety guarantee or as evidence of broad real-world safety improvement.
> > >
> > > ### 7. Correcting the LLM Usage Statement
> > >
> > > We agree that the previous statement was incomplete and potentially misleading. The revised **“Usage of LLMs”** section now clearly distinguishes the two uses of LLMs.
> > >
> > > First, **GPT-4 is a core high-level planning component** of ChatAni. It is prompted as Oracle and Actor agents to:
> > >
> > > - parse user instructions;
> > > - assign participant roles;
> > > - infer interaction groups and behaviors;
> > > - generate schedules and keypoint instructions;
> > > - return JSON-formatted structured outputs, with selected text fields containing natural-language behavior descriptions.
> > >
> > > Deterministic tools convert these outputs into plans executed by PedAnimator and VehAnimator. GPT-4 is not used to train the low-level animation policies or evaluate generated results.
> > >
> > > Second, LLM-based tools may have been used for minor writing polishing, with the technical content and claims checked by the authors.
> > >
> > > We agree that broader evaluation across alternative LLM backbones would be valuable. The appendix includes a limited comparison among **GPT-3.5, Llama-3 70B, and GPT-4**, showing that the planner is not evaluated only with a single backbone. However, this experiment is not a systematic study of more recent models, a wider range of model scales, or prompt sensitivity. We therefore do not claim that the reported planner performance is independent of GPT-4, and the revised limitations explicitly state both the reliance on a **proprietary model** in the main system and the remaining need for a broader backbone study.

---

> > > > ### Author Response · Authors · 2026-07-19
> > > > **Response to Reviewer dcqR(4)**
> > > >
> > > > ### 8. Reproducibility and Artifact Availability
> > > >
> > > > We revise the reproducibility statement to describe the currently provided artifacts precisely rather than claiming that every artifact required for full reproduction has already been publicly packaged.
> > > >
> > > > The current supplementary materials contain:
> > > >
> > > > - **training code and configuration files for the main experiments**;
> > > > - **selected evaluation code**;
> > > > - the high-level planning design and implementation;
> > > > - prompt templates;
> > > > - model settings;
> > > > - pseudocode;
> > > > - detailed evaluation protocols.
> > > >
> > > > Because of the limited revision period and the engineering effort required to clean, validate, document, and package the entire research codebase and large artifacts to a public-release standard, we were unable to complete the full release before the current submission deadline. This limitation concerns the time required for release preparation rather than the absence of implementation materials. To provide concrete support for reproducibility in the current submission, we include the available training code and configurations for the main experiments, selected evaluation code, high-level planning implementation and prompts, model settings, pseudocode, and detailed protocols. These materials allow the main implementation choices and experimental procedures to be inspected and provide substantive evidence supporting reproducibility of the reported work, although we do not describe the current artifact package as fully complete.
> > > >
> > > > After completing this curation and documentation process, we will release the **full codebase, generated data, and trained checkpoints** through a public repository, subject to storage constraints and third-party dataset licenses. The revised manuscript clearly distinguishes the artifacts available now from this planned complete release and does not describe the current artifact package as fully complete.
> > > >
> > > > ### 9. Stronger and Better-Scoped Baseline Comparisons
> > > >
> > > > We add two recent and more closely related baselines:
> > > >
> > > > - **UniPhys** for physics-based character control;
> > > > - **Talk2Traffic** for language-conditioned traffic-scenario generation.
> > > >
> > > > UniPhys is added to the PedAnimator trajectory-following and motion-imitation comparison. Talk2Traffic is added to the high-level planning comparison.
> > > >
> > > > We also clarify why no single existing method supports the complete ChatAni setting, which combines language-conditioned planning, map-constrained multi-actor traffic control, physics-based pedestrian animation, kinematics-aware vehicle animation, and pedestrian/vehicle interactions. We therefore adapt the closest available baselines to our scenarios and shared interfaces while retaining their core models, and compare only the metrics each method can meaningfully support:
> > > >
> > > > - Pacer, Pacer+, and UniPhys for pedestrian animation;
> > > > - Pure Pursuit and the learning-based vehicle controller for vehicle tracking;
> > > > - LCTGen, ChatSim, and Talk2Traffic for language-conditioned planning.
> > > >
> > > > The paper now explicitly states that UniPhys does not directly support traffic-map or pedestrian-vehicle interaction tasks, while Talk2Traffic produces Scenic/CARLA-style scenarios rather than low-level Waymo-map-based animation plans. Comparisons are restricted to metrics supported by each baseline.
> > > >
> > > > ### 10. Expanded Limitations
> > > >
> > > > We expand the limitations section to state the current boundaries of ChatAni more directly:
> > > >
> > > > - ChatAni currently supports pedestrians and vehicles but not **cyclists or cyclist-specific dynamics**.
> > > > - Executable behaviors remain constrained by the supported behavior categories, controller objectives, and animation primitives; the system is not fully open-vocabulary at the execution level.
> > > > - The planner relies on **proprietary GPT-4** in the main system. Although the appendix includes a limited GPT-3.5/Llama-3 70B/GPT-4 comparison, more recent backbones, a wider range of model scales, and prompt sensitivity have not been systematically evaluated.
> > > > - Conservative collision avoidance can produce **command-inconsistent vehicle plans** when traffic blocks the requested maneuver.
> > > > - PedAnimator may prioritize balance over motion fidelity for actions with large center-of-mass changes.
> > > > - Pedestrian-vehicle collisions do not model high-fidelity impact response, body deformation, detailed contact geometry, or post-impact recovery.
> > > > - The downstream augmentation experiments use a **single seed**, and the DriveLM evaluation is restricted to generated hazardous scenes.
> > > >
> > > > We believe these revisions provide a more accurate and balanced account of both the demonstrated capabilities and the remaining limitations of ChatAni.